# Is Graph Unlearning Ready for Practice? A Benchmark on Efficiency, Utility, and Forgetting

**Samyak Jain**[1*]  **Ronak Kalvani**[2*]  **Sainyam Galhotra**[4,5]  **Sayan Ranu**[1,3]

[1]Department of Computer Science and Engineering  [2]Department of Mathematics and Computing
[3]Yardi School of Artificial Intelligence  [4]Department of Computer Science
Indian Institute of Technology Delhi, New Delhi, 110016, India  [5]Cornell University
`samyakjain1729@gmail.com`, {`mt6210952@`, `sayanranu@cse`}`.iitd.ac.in`, `sg@cs.cornell.edu`

## Abstract

Graph Neural Networks (GNNs) are increasingly being deployed in sensitive, user-centric applications where regulations such as the GDPR mandate the ability to remove data upon request. This has spurred interest in graph unlearning, the task of removing the influence of specific training data from a trained GNN without retraining from scratch. While several unlearning techniques have recently emerged, the field lacks a principled benchmark to assess whether these methods truly provide a practical alternative to retraining and, if so, how to choose among them for different workloads. In this work, we present the first systematic benchmark for GNN unlearning, structured around three core desiderata: *efficiency* (is unlearning faster than retraining?), *utility* (does the unlearned model preserve predictive performance and align with the retrained gold standard?), and *forgetting* (does the model genuinely eliminate the influence of removed data?). Through extensive experiments across diverse datasets and deletion scenarios, we deliver a unified assessment of existing approaches, surfacing their trade-offs and limitations. Crucially, our findings show that most unlearning techniques are not yet practical for large-scale graphs. At the same time, our benchmarking yields actionable guidelines on when unlearning can be a viable alternative to retraining and how to select among methods for different workloads, thereby charting a path for future research toward more practical, scalable, and trustworthy graph unlearning.

## 1 Introduction

The ability to remove the influence of specific training data from machine learning models, commonly referred to as *machine unlearning*, is increasingly critical for real-world applications (Bourtoule et al., 2021; Cao and Yang, 2015). Requests for data deletion arise in practice due to evolving privacy regulations, such as the General Data Protection Regulation (GDPR) (Mantelero, 2013), which grants individuals the "right to be forgotten" by requiring organizations to erase personal data upon request. Compliance with such regulations is not optional: organizations face substantial legal and financial penalties for failure to comply, making efficient unlearning a practical necessity.

A growing body of work has investigated algorithmic frameworks for GNN unlearning (Cheng et al., 2023; Li et al., 2024; Wu et al., 2023; Dong et al., 2024; Pan et al., 2023; Chen et al., 2022; Wang et al., 2023). Collectively, these works underscore the importance of graph unlearning and have established it as an active area of research.

Despite notable progress, the field still lacks a principled benchmarking framework capable of addressing two fundamental questions: *(1) Do existing GNN unlearning methods offer a tangible advantage over retraining from scratch on the modified dataset? (2) If so, how can practitioners determine the most suitable unlearning algorithm for a given workload?* Answering these questions remains elusive because current evaluations suffer from the following key gaps:

- **Evaluation ambiguity:** Should unlearning aim to replicate the predictive performance of the gold standard (e.g., AUC, F1-score), the similarity of output logits, or the closeness of model parameters? Current works inconsistently adopt these metrics, leading to incomparable results and potentially misleading conclusions.

---

*Denotes equal contribution.

- **Efficiency vs. retraining:** For unlearning to be practically useful, it must be faster than retraining from scratch after data deletion. If an unlearning method takes longer than retraining, or exceeds GPU memory capacity, it becomes redundant regardless of its effectiveness. Yet, systematic efficiency comparisons against retraining on large datasets are frequently missing in existing studies.

- **Generalization and robustness:** The difficulty of unlearning vary with graph structure (e.g., homophilous vs. heterophilous networks), deletion scenarios (e.g., random removals vs. targeted deletions within specific communities), and GNN architectures. Yet, this diversity remains under-explored, leaving open questions about robustness and generalization. Moreover, many methods that provide "certified" unlearning rely on simplifying assumptions, such as restricting to linear GNNs (Guo et al., 2023; Yi and Wei, 2025). In practice, however, non-linear GNNs are far more common, and their behavior under unlearning is less understood.

To address the above gaps, our benchmarking study makes the following key contributions:

1. **Comprehensive and diagnostic benchmarking framework.** While prior unlearning papers evaluate some subset of utility, efficiency, and forgetting, our contribution lies in conducting the *first holistic, multi-layered diagnostic assessment* of GNN unlearning. Concretely, we evaluate: (i) *utility* across three complementary axes—aggregate metrics (accuracy, fidelity), per-instance outputs (logit-space deviations), and parameter-space proximity—revealing discrepancies missed by prior single-metric evaluations; (ii) *forgetting quality* using a broader suite of privacy attacks, including membership inference Olatunji et al. (2021), unlearning inversionZhang et al. (2026), and noisy-labeler attacks Sui et al. (2025); (iii) *efficiency* through both runtime and memory profiling, explicitly accounting for preprocessing overheads that prior work omits; and (iv) *robustness* to realistic and adversarial deletion distributions (e.g., degree-based, label-skewed), going beyond the uniform-random setups that dominate existing evaluations. This expanded scope enables a more reliable understanding of when unlearning works, when it fails, and why.

2. **Actionable insights and decision support.** Our empirical analysis surfaces critical trade-offs across efficiency, utility, and forgetting, showing that while unlearning can sometimes approximate retraining well, in many practical settings, retraining-from-scratch remains the most compelling option. Our results call for a re-orientation of research priorities: paradigms that emphasize theoretical guarantees under narrow assumptions (e.g., projection or certification-based methods) offer limited practical value, while the most promising techniques must be extended with scalable, batchable implementations if they are to remain competitive with retraining.

3. **Open-source implementation.** We provide a rich, open-source codebase covering multiple state-of-the-art graph unlearning algorithms, ensuring reproducibility and enabling future work to build upon a common and fair experimental ground (`https://github.com/idea-iitd/Unlearning_Benchmark`).

At this juncture, we note the existence of a benchmarking effort on GNN unlearning (Fan et al., 2025). Our study differs in three fundamental ways. First, it goes beyond raw experimentation to provide actionable guidance: we distill empirical findings into simple principles that practitioners can use to decide when unlearning is viable and when retraining is the only realistic option. Second, unlike prior work, which measures efficacy only through aggregate metrics (e.g., AUC-ROC), we assess alignment at multiple levels spanning prediction fidelity, logit distributions, and weight space, and thereby capturing subtler divergences from retraining. Third, we foreground efficiency relative to retraining as an indispensable criterion: an unlearning method slower than retraining offers little practical value, yet this baseline is overlooked. Finally, we move beyond the unrealistic assumption that deletions are uniformly random, and evaluate robustness to a wider array of disitributions.

## 2 PRELIMINARIES

**Definition 2.1** (Graph). *A graph is represented as $\mathcal{G} = (\mathcal{V}, \mathcal{E}, \mathcal{X})$, where $\mathcal{V}$ is a finite set of vertices (or nodes), $\mathcal{E} \subseteq \mathcal{V} \times \mathcal{V}$ is a set of edges, and $\mathcal{X} \in \mathbb{R}^{|\mathcal{V}| \times d}$ is a feature matrix, where each row $\mathbf{x}_v \in \mathbb{R}^d$ corresponds to the $d$-dimensional feature vector of node $v \in \mathcal{V}$.*

We study graph unlearning in the context of the *node classification*.

**Definition 2.2** (Node Classification with GNNs ). *Given a graph $\mathcal{G} = (\mathcal{V}, \mathcal{E}, \mathcal{X})$, let $\mathcal{V}_{train} \subseteq \mathcal{V}$ denote the set of training nodes with associated labels $y_{train}$. The learning task is to infer labels for the remaining nodes $\mathcal{V} \setminus \mathcal{V}_{train}$ under a specified evaluation regime $T \in \{transductive, inductive\}$. A GNN $\mathcal{A}_{GNN}$ takes as input the dataset $D = (\mathcal{V}_{train}, \mathcal{E}, \mathcal{X}, y_{train})$ and outputs model parameters $\Theta \leftarrow \mathcal{A}_{GNN}(D)$. The trained predictor is denoted by $f_\Theta$, which maps nodes in $\mathcal{V}$ to label distributions.*

**Definition 2.3** (Graph Unlearning). *Let $\mathcal{G} = (\mathcal{V}, \mathcal{E}, \mathcal{X})$ be a graph with training dataset $D = (\mathcal{V}_{train}, \mathcal{E}, \mathcal{X}, y_{train})$. Consider a subset of training nodes $\Delta \mathcal{V}_{train} \subseteq \mathcal{V}_{train}$ that must be removed. The modified training dataset is then $D' = (\mathcal{V}_{train} \setminus \Delta \mathcal{V}_{train}, \mathcal{E}, \mathcal{X}, y'_{train})$, where $y'_{train}$ are labels of nodes $\mathcal{V}_{train} \setminus \Delta \mathcal{V}_{train}$.*

*Denote by $\Theta' \leftarrow \mathcal{A}_{\text{GNN}}(D')$ the parameters obtained by retraining a GNN from scratch on $D'$. A graph unlearning algorithm $\mathcal{U}$ takes as input the original parameters $\Theta$ (trained on $D$) together with the removal set $\Delta \mathcal{V}_{train}$ and produces updated parameters $\tilde{\Theta} \leftarrow \mathcal{U}(\Theta, \Delta \mathcal{V}_{train})$. The objective of unlearning is twofold: (i) the unlearned model $\tilde{\Theta}$ should be similar the gold model $\Theta'$ on the retained data, and (ii) it should forget the influence of the removed set $\Delta \mathcal{V}_{\text{train}}$.*

**Model similarity:** No consensus exists on how to measure the similarity between the gold standard model $\Theta'$ and the unlearned model $\tilde{\Theta}$. The similarity may be assessed at three different layers, each with its own advantages and limitations:

- **Parameter space similarity.** Here, similarity is defined directly between the parameter sets $\Theta'$ and $\tilde{\Theta}$, for instance, using norm-based distances such as $\ell_2$ or cosine distance. If the parameters are close, then the intermediate activations at each GNN layer are also likely to be similar (See Lemma A.1), thereby providing a strong indication that the unlearned model closely matches the retrained one in terms of the functions these two models have learned.
- **Logit space similarity.** The mapping from parameters to outputs is not one-to-one since two different parameterizations may produce similar predictions. Hence, an alternative view is to focus on the distributions produced by the two predictors. For a node $v$ with feature $\mathbf{x}_v$, $f_{\Theta'}(\mathbf{x}_v)$ and $f_{\tilde{\Theta}}(\mathbf{x}_v)$ define probability distributions over the label space. Distances between these distributions can be quantified using measures such as KL-divergence, Jensen–Shannon divergence, Earth Mover's Distance (EMD), or $\ell_p$ distances.
- **Similarity in utility space.** Different logit distributions may still produce similar outputs due to the discretization of the continuous logit space. Hence, the coarsest view compares task-level metrics such as training/validation loss, AUCROC, or F1-scores obtained by $f_{\Theta'}$ and $f_{\tilde{\Theta}}$.

**Forgetting:** Two models may exhibit high similarity and yet differ significantly in the amount of information they retain about removed nodes. Hence, forgetting must be assessed separately from utility. Just like model similarity, there is no single all-encompassing measure to quantify the extent of forgetting. We use the three-pronged strategy to evaluate forgetting: (i) membership information of removed nodes in the original training set, (ii) structural leakage caused by message passing, and (iii) confidence patterns correlated with the removed training signals. The technical details of how these aspects are assessed are discussed in § 4.

## 3 GNN UNLEARNING: THE CURRENT LANDSCAPE

Table 1 characterizes existing techniques for GNN unlearning across multiple dimensions. Below, we unpack each dimension.

**Unlearning Paradigm.** Existing research on GNN unlearning spans several overlapping methodological directions, each grounded in different principles for approximating retraining:

- **Learning-based approaches** directly update or fine-tune model parameters to mimic the effect of removing training data. Examples include GNNDELETE (Cheng et al., 2023) and MEGU (Li et al., 2024), which design efficient update rules or gradient-based corrections to balance unlearning effectiveness with predictive accuracy.
- **Influence-function and projection** approaches analytically approximate the effect of removing data. Influence-function methods such as GIF (Wu et al., 2023), IDEA (Dong et al., 2024), ETR (Yang et al., 2025) and GST (Pan et al., 2023) estimate how the loss landscape would change if a sample were deleted and then adjust parameters accordingly. Projector (Cong and Mahdavi, 2023) extends this idea by restricting updates to subspaces aligned with retraining dynamics, thereby offering guarantees of exact unlearning. However, PROJECTOR *is tied to a narrowly defined, custom-built linear* GNN *and lacks the flexibility to operate on general* GNN *architectures.*

- **Partition-based approaches (SISA-style)** such as GRAPHERASER (Chen et al., 2022) and GUIDE (Wang et al., 2023) follow a shard-and-aggregate paradigm: the graph is partitioned into smaller subgraphs (shards), and unlearning is applied only to the affected shards before aggregating results. While this design can be computationally efficient, it is highly sensitive to partition quality and does not reflect real-world graphs that are rarely shardable in a natural way. Moreover,

*the guarantees of exactness provided by these methods hold only under the restrictive assumption that a separate* GNN *is trained per shard.* If a single GNN is trained over the full graph, these guarantees break down.

- **Certified approaches.** A distinct line of work, exemplified by Certified Graph Unlearning (Chien et al., 2022) and SCALEGUN (Yi and Wei, 2025), provides formal guarantees on forgetting. However, these methods are restricted to a very narrow setting. Specifically, they operate only on custom-designed linear GNNs because their influence-function-based updates rely on tractable gradients and Hessians, which remain low-rank and invertible only under linear message passing. This restriction prevents them from being applied to widely other non-linear GNNs , and renders non-convex optimizations inapplicable, as doing so would invalidate the theoretical guarantees. Additionally, these methods are limited to one-vs-rest (binary) classification: each class has an independent binary Hessian, whereas multiclass softmax couples parameters across classes, producing a dense Hessian that breaks the efficient update scheme. In effect, the pursuit of formal guarantees comes at the cost of practical flexibility and applicability to real-world GNN settings. Non linear methods like (Dong et al., 2024) inject noise (like gaussian noise) which lead to large error bounds on larger graphs where unlearning is crucially needed as retraining from scratch is hard. This limits utility on large graph.

**Property.** Columns 3–6 describe four important aspects of how unlearning is operationalized: *Model Agnostic* methods generalize across architectures and hence are more versatile. *Continuous Training* indicates the ability to absorb new training data following a round of unlearning. Techniques that lack this capability are restricted to static snapshots only and hence are not deployment friendly. This limitation primarily arises when an unlearning algorithm adds additional neural or transformation lay-

Table 1: Comparison of graph unlearning methods across multiple dimensions (GNNDELETE(Cheng et al., 2023), MEGU(Li et al., 2024), GIF(Wu et al., 2023), IDEA(Dong et al., 2024), GST(Pan et al., 2023), PROJECTOR(Cong and Mahdavi, 2023), GRAPHERASER(Chen et al., 2022), GUIDE(Wang et al., 2023), SCALEGUN (Yi and Wei, 2025), CGU (Chien et al., 2022), COGNAC (Kolipaka et al., 2024), ETR (Yang et al., 2025)). This study covers peer-reviewed unlearning algorithms only. IF indicates Influence-Function.

| Technique | Paradigm | Properties | | | |
|---|---|---|---|---|---|
| | | Model Ag. | Cont. Train. | Mode | Guarantee |
| GNNDELETE | Learning | ✓ | ✗ | Post-hoc | ✗ |
| MEGU | Learning | ✓ | ✓ | Post-hoc | ✗ |
| GIF | IF | ✓ | ✗ | Post-hoc | ✗ |
| IDEA | IF, certified | ✓ | ✗ | Post-hoc | ✗ |
| GST | IF | ✓ | ✓ | Post-hoc | ✗ |
| PROJECTOR | Projection | ✗ | ✗ | Train-time | ✓ |
| GRAPHERASER | SISA | ✓ | ✓ | Train-time | ✓ |
| GUIDE | SISA | ✓ | ✓ | Train-time | ✓ |
| Certified GU | Certified | ✗ | ✓ | Train-time | ✓ |
| SCALEGUN | Certified | ✗ | ✓ | Train-time | ✓ |
| COGNAC | Corrective | ✓ | ✓ | Post-hoc | ✗ |
| ETR | IF, Learning | ✓ | ✓ | Post-hoc | ✗ |

ers to the original GNN to unlearn. The *Mode* column denotes whether unlearning is applied *post-hoc* (to a pre-trained model) or must be integrated into *train-time* procedures (e.g., via partitioning or certification). This category also includes techniques that assume a tailored GNN architecture for their unlearning method to work. Post-hoc methods are more flexible, while train-time methods provide stronger guarantees at the cost of higher training complexity. Finally, a subset of techniques provides a *guarantee of exactness* or *certification*. However, such guarantees invariably rely on restrictive assumptions as discussed.

## 4 BENCHMARKING FRAMEWORK

An effective unlearning algorithm must satisfy three fundamental constraints:

- **Efficiency.** The algorithm should be computationally more efficient than retraining the model from scratch on the updated training data (the gold standard).

- **Model utility.** The predictive performance of the unlearned model should remain comparable to that of the gold standard. This ensures that the removal of data does not lead to disproportionate degradation in downstream tasks, making the model practically useful after unlearning.

- **Forgetting quality.** An effective unlearning algorithm must ensure that the influence of the removed data is genuinely eliminated from the model. This requirement is often mandated by legal frameworks such as GDPR or driven by user demands for data erasure.

We next outline the metrics used to assess the extent to which these constraints are satisfied.

Table 2: Statistical overview of benchmark datasets.

| Dataset | Nodes | Edges | Features | Classes | Type | Description |
|---|---|---|---|---|---|---|
| CORA | 2,708 | 5,278 | 1,433 | 7 | Homophily | Citation Network |
| CITESEER | 3,327 | 4,732 | 3,703 | 6 | Homophily | Citation Network |
| OGBN-Arxiv | 169,343 | 1,166,243 | 128 | 40 | Homophily | Citation Network |
| PHOTO | 7,487 | 119,043 | 745 | 8 | Homophily | Co-purchasing Network |
| AMAZON-Ratings | 24,492 | 93,050 | 300 | 5 | Heterophily | Rating Network |
| ROMAN-Empire | 22,662 | 32,927 | 300 | 18 | Heterophily | Article Syntax Network |
| Reddit | 232,965 | 114,615,892 | 602 | 41 | Homophily | Social Network |

**Efficiency.** The efficiency of an unlearning algorithm is measured by its running time relative to retraining the gold-standard model. An algorithm is deemed practical only if it offers a tangible speedup over full retraining on the updated dataset.

**Utility.** Model utility is assessed at multiple levels:
- *Aggregate accuracy:* The percentage of correctly classified nodes by model $\Theta$, i.e.,

$$\text{Acc} = \frac{1}{|V_{\text{test}}|} \sum_{v \in V_{\text{test}}} \mathbb{1}[f_\Theta(v) = y_v]. \tag{1}$$

- *Per-instance similarity:* At the granularity of individual predictions, we evaluate:
  - *Fidelity:* The fraction of test nodes for which the unlearned model $\tilde{\Theta}$ makes the same prediction as the gold-standard model $\Theta'$,

$$\text{Fidelity} = \frac{1}{|V_{\text{test}}|} \sum_{v \in V_{\text{test}}} \mathbb{1}[f_{\tilde{\Theta}}(v) = f_{\Theta'}(v)]. \tag{2}$$

  - *Logit-level distance:* The $\ell_2$ distance between the predicted probability distributions (logits) of the unlearned and gold-standard models, averaged across all test nodes.

  Fidelity and logit distance are important because two models can achieve nearly identical accuracy while making very different individual predictions. Thus, fidelity and logit distance capture per-instance agreement that aggregate metrics like accuracy alone may obscure.

- **Model-level similarity.** Finally, we compare the parameters of the unlearned model $\tilde{\Theta}$ and the gold-standard model $\Theta'$. Similarity in the parameter space is measured via the $\ell_2$ distance:

**Forgetting** To quantify forgetting, we adopt three complementary families of attacks, each probing a different form of residual influence from the removed set.

- **Membership Inference Attacks (MIA).** These evaluate whether an adversary can determine if a node belonged to the original training set. MIAs test a basic privacy requirement, but capture only one dimension of forgetting and cannot detect structural or label-level leakage. For more details on the mathematical formulation, please refer to App. A.2.

- **Unlearning Inversion Attacks (UIA) (Zhang et al., 2026).** Inversion-based attacks attempt to reconstruct *deleted edges* from black-box access to $\tilde{\Theta}$. These attacks detect structural leakage caused by message passing and confidence perturbations, which MIAs do not capture.

- **Noisy-Labeler Attacks (Sui et al., 2025).** We additionally evaluate whether $\tilde{\Theta}$ acts as a "noisy labeler," assigning removed nodes high-confidence predictions that reflect their original class memberships. This probes *label leakage*, complementing MIAs and structural leakage (UIA).

## 5 BENCHMARKING EVALUATION

As discussed in § 1, our central question is when can we use an existing unlearning algorithm or should we simply retrain. Building on the three pillars of *efficiency*, *utility*, and accurate *forgetting* of removed data, we study the following research questions (RQs):
- **RQ1. Utility.** To what extent does an unlearned model approximate the distribution of a model retrained from scratch, and under what circumstances does this approximation break down?
- **RQ2. Efficiency.** Do existing unlearning techniques actually offer computational savings (time and memory) over retraining, or does retraining remain the practical baseline?
- **RQ3. Forgetting.** Do existing unlearning algorithms successfully forget the deleted data?
- **RQ4. Large-scale unlearning.** Can methods process multiple successive deletion requests without catastrophic loss in utility or efficiency?
- **RQ5. Robustness across workloads.** How stable are unlearning methods when deletion requests come from non-random distributions (high degrees, label frequency, etc.)?

While we primarily focus on unlearning over node deletions, in App. A.3 and App. A.4, we also evaluate performance edge and feature unlearning.

Table 3: Accuracies for unlearning 10% nodes (higher is better). Best results are shaded.

| Technique | CORA | CITESEER | PHOTO | AMAZON-R. | ROMAN-E. | OGBN-ARXIV |
|---|---|---|---|---|---|---|
| GOLD (GCN) | 0.88 ± 0.00 | 0.73 ± 0.01 | 0.93 ± 0.00 | 0.43 ± 0.01 | 0.41 ± 0.00 | 0.60 ± 0.01 |
| Original Acc (GCN) | 0.89 ± 0.00 | 0.77 ± 0.00 | 0.92 ± 0.00 | 0.43 ± 0.00 | 0.41 ± 0.00 | 0.61 ± 0.00 |
| GOLD (PROJECTOR) | 0.84 ± 0.00 | 0.77 ± 0.00 | 0.87 ± 0.00 | 0.47 ± 0.00 | 0.50 ± 0.00 | 0.61 ± 0.00 |
| Original Acc (PROJECTOR) | 0.84 ± 0.00 | 0.76 ± 0.00 | 0.87 ± 0.00 | 0.47 ± 0.00 | 0.50 ± 0.00 | 0.61 ± 0.00 |
| MEGU | 0.89 ± 0.00 | 0.77 ± 0.00 | 0.92 ± 0.00 | 0.42 ± 0.00 | 0.41 ± 0.00 | 0.61 ± 0.00 |
| GIF | 0.88 ± 0.00 | 0.76 ± 0.00 | 0.92 ± 0.00 | 0.43 ± 0.00 | 0.44 ± 0.00 | 0.59 ± 0.00 |
| IDEA | 0.87 ± 0.00 | 0.77 ± 0.00 | 0.93 ± 0.00 | 0.41 ± 0.00 | 0.48 ± 0.00 | 0.56 ± 0.00 |
| PROJECTOR | 0.84 ± 0.00 | 0.77 ± 0.00 | 0.87 ± 0.00 | 0.47 ± 0.00 | 0.50 ± 0.00 | 0.61 ± 0.00 |
| GRAPHERASER | 0.85 ± 0.00 | 0.76 ± 0.00 | 0.92 ± 0.00 | 0.42 ± 0.00 | 0.34 ± 0.00 | 0.59 ± 0.00 |
| GNNDELETE | 0.76 ± 0.05 | 0.76 ± 0.01 | 0.34 ± 0.11 | 0.37 ± 0.00 | 0.33 ± 0.00 | OOM |
| GST | OOM | OOM | OOM | OOM | OOM | OOM |
| GUIDE | 0.84 ± 0.00 | 0.71 ± 0.05 | 0.85 ± 0.00 | 0.23 ± 0.01 | 0.26 ± 0.01 | OOM |
| COGNAC | 0.84± 0.00 | 0.68± 0.05 | 0.92± 0.01 | 0.45± 0.01 | 0.51 ± 0.00 | 0.68± 0.00 |
| ETR | 0.89± 0.00 | 0.81± 0.05 | 0.93± 0.01 | 0.41± 0.01 | 0.30 ± 0.00 | 0.58± 0.00 |

## 5.1 EXPERIMENTAL SETUP

App. A.1 outlines our hardware/software environment, hyperparameters, and codebase. Each experiment is repeated five times, and we report the mean and standard deviation. Dataset statistics are provided in Table 2. The Reddit dataset is used primarily to test scalability. By default, deletion workloads consist of nodes sampled uniformly at random unless noted otherwise. OOT indicates that an algorithm failed to complete with 24 hours. OOM, on the other hand, indicates exceeding the GPU memory bandwidth (A100, 40GB).

**Benchmarked algorithms:** We benchmark all algorithms listed in Table 1 except the certification-based methods, namely SCALEGUN (Yi and Wei, 2025) and CGU (Chien et al., 2022). These two techniques are excluded from the main benchmarking as they are limited to one-Vs.-all binary classification and restricted to a specific class of linear GNNs. Nonetheless, for completeness, these two techniques are compared separately, and the results are presented in App. A.5.

For all model-agnostic unlearning algorithms, unless otherwise specified, we use GCN as the default backbone. PROJECTOR is also restricted to linear GNNs, but unlike certification-based algorithms, it generalizes to multi-class classification. We therefore include PROJECTOR in our benchmarks for completeness, while emphasizing that its results should be interpreted with caution due to the architectural mismatch.

## 5.2 RQ1. EVALUATING UTILITY

To address **RQ1**, we systematically compare the unlearning performance of existing techniques across multiple datasets by evaluating the changes in model behavior at four hierarchical levels of the information space: (i) prediction accuracy, (ii) output fidelity through similarity of prediction distributions, (iii) representation-level differences in the logit space, and (iv) parameter-level changes. First, we analyze how the different techniques perform on these metrics at a fixed unlearning ratio. Next, we examine how varying the unlearning ratio affects their overall effectiveness.

**Accuracy.** In Table 3, we see that for homophilous datasets, such as CORA, CITESEER and PHOTO, the accuracy of GOLD is closely matched by MEGU, GIF, and IDEA. However, the relative performance ranking shifts when moving to heterophilous datasets such as AMAZON-Ratings and ROMAN-Empire. Here, PROJECTOR has the highest accuracy, thanks to its customized linear GNN doing better than the GCN used by the rest of the algorithms. GST runs out of memory across all datasets. Hence, we omit this algorithm from further benchmarking.

**Fidelity.** Two models can achieve similar accuracy yet diverge in fidelity, meaning one is not truly aligned with the other. Table 4 reports the fidelity results. PROJECTOR achieves the highest fidelity overall, but this is relative to its own customized linear GNN, making the comparison with other methods less direct. The fact that PROJECTOR can exhibit lower accuracy yet superior fidelity highlights that accuracy alone is an unreliable metric for assessing unlearning quality. Among GCN-based approaches, MEGU consistently achieves the strongest fidelity on average, underscoring its robustness across datasets.

**Logit distance.** Table 5 reports pairwise $\ell_2$ distances between the logits of unlearned models and the GOLD retrained models, providing a finer-grained view of distributional similarity. Unlike fidelity,

Table 4: Fidelity at an unlearn ratio of 0.1 (higher is better). Projector achieves the highest fidelity (highlighted in gold); however, it operates on a customized linear GNN specifically designed to enable 'exact' unlearning, making direct comparisons unfair. To provide a fair reference, we also highlight in gray the best fidelity achieved by other methods when applied to the standard GCN.

| Technique | CORA | CITESEER | PHOTO | AMAZON-R. | ROMAN-E. | Ogbn-arxiv |
|---|---|---|---|---|---|---|
| MEGU | $0.95 \pm 0.00$ | $0.85 \pm 0.01$ | $0.99 \pm 0.00$ | $0.84 \pm 0.03$ | $0.86 \pm 0.00$ | $0.91 \pm 0.01$ |
| GIF | $0.93 \pm 0.01$ | $0.82 \pm 0.01$ | $0.97 \pm 0.00$ | $0.87 \pm 0.01$ | $0.57 \pm 0.00$ | $0.85 \pm 0.00$ |
| IDEA | $0.92 \pm 0.00$ | $0.83 \pm 0.01$ | $0.95 \pm 0.00$ | $0.82 \pm 0.01$ | $0.53 \pm 0.00$ | $0.76 \pm 0.00$ |
| PROJECTOR | $0.99 \pm 0.00$ | $1.00 \pm 0.00$ | $0.99 \pm 0.00$ | $0.99 \pm 0.00$ | $0.98 \pm 0.00$ | $0.98 \pm 0.00$ |
| GRAPHERASER | $0.88 \pm 0.00$ | $0.81 \pm 0.01$ | $0.98 \pm 0.00$ | $0.85 \pm 0.01$ | $0.60 \pm 0.01$ | $0.90 \pm 0.01$ |
| GNNDELETE | $0.79 \pm 0.06$ | $0.83 \pm 0.01$ | $0.35 \pm 0.12$ | $0.67 \pm 0.01$ | $0.62 \pm 0.01$ | OOM |
| GUIDE | $0.84 \pm 0.004$ | $0.79 \pm 0.00$ | $0.96 \pm 0.003$ | $0.81 \pm 0.01$ | $0.70 \pm 0.02$ | OOM |
| Cognac | $0.90 \pm 0.01$ | $0.87 \pm 0.01$ | $0.97 \pm 0.01$ | $0.81 \pm 0.02$ | $0.65 \pm 0.01$ | $0.81 \pm 0.01$ |
| ETR | $0.88 \pm 0.01$ | $0.82 \pm 0.01$ | $0.96 \pm 0.01$ | $0.81 \pm 0.02$ | $0.48 \pm 0.01$ | $0.83 \pm 0.01$ |

Table 5: Pairwise Logit L2 Distance (GOLD vs. Techniques) at an Unlearn Ratio of 0.1. Lower is better. The color scheme is explained in Table 4 caption.

| Technique | CORA | CITESEER | PHOTO | AMAZON-R. | ROMAN-E. | OGBN-ARXIV |
|---|---|---|---|---|---|---|
| MEGU | $3.05 \pm 0.20$ | $6.31 \pm 0.23$ | $3.81 \pm 0.68$ | $0.59 \pm 0.15$ | $1.57 \pm 0.24$ | $3.53 \pm 0.12$ |
| GIF | $3.60 \pm 0.19$ | $6.33 \pm 0.23$ | $4.20 \pm 0.61$ | $0.56 \pm 0.12$ | $5.24 \pm 0.13$ | $5.03 \pm 0.14$ |
| IDEA | $18.03 \pm 0.80$ | $10.06 \pm 0.44$ | $22.79 \pm 1.88$ | $4.69 \pm 0.19$ | $22.62 \pm 0.64$ | $45.79 \pm 0.63$ |
| PROJECTOR | $0.09 \pm 0.00$ | $0.08 \pm 0.00$ | $0.27 \pm 0.01$ | $0.17 \pm 0.00$ | $0.42 \pm 0.01$ | $0.41 \pm 0.00$ |
| GNNDELETE | $5.91 \pm 0.26$ | $7.36 \pm 0.19$ | $8.57 \pm 0.46$ | $1.10 \pm 0.06$ | $2.64 \pm 0.16$ | OOM |
| Cognac | $19.7 \pm 0.39$ | $27.49 \pm 0.28$ | $5.8958 \pm 0.89$ | $0.63 \pm 0.19$ | $9.13 \pm 0.89$ | $11.28 \pm 0.93$ |
| ETR | $17.11 \pm 0.39$ | $32.89 \pm 0.25$ | $5.9358 \pm 0.87$ | $1.02 \pm 0.19$ | $5.38 \pm 0.89$ | $7.63 \pm 0.93$ |

which is affected by the thresholding of logits into discrete predictions, logit distances capture discrepancies at the level of the entire output distribution. This makes them a more sensitive diagnostic for alignment with GOLD. Note that this comparison is not meaningful for SISA-based approaches, since these techniques aggregate predictions across shards rather than using a single model. As a result, we omit their logit-level evaluation.

Consistent with the fidelity results, PROJECTOR and MEGU achieves the closest alignments to gold. By contrast, IDEA exhibits substantial divergence in logit space, despite achieving reasonably strong fidelity in Table 4. This discrepancy indicates that while IDEA often predicts the same labels as GOLD, it does so with very different confidence distributions.

**Model-level distance.** This analysis is only feasible for methods that preserve the original GNN architecture after unlearning. Approaches that alter the architecture, e.g., by introducing additional neural layers are not considered, since their parameters are no longer in the same space as the GOLD model. Likewise, sharding-based methods are also excluded because they train multiple models and aggregate their predictions, making a direct parameter-level comparison inconsistent with their design. These constraints leave us with MEGU, GIF, and IDEA.

Table 6 reports the relative L2 distances in weight space. At an unlearning ratio of $0.1$, the methods are virtually indistinguishable. As the workload increases to $0.8$, minor separations emerge, with MEGU showing a slight advantage. We further note that GIF and IDEA yield identical distances, since they share the same training pipeline, with differences arising only during inference.

Table 6: L2 distance between model parameters with GOLD at unlearn ratios $0.1$ and $0.8$. Lower is better.

| Dataset | GOLD_vs_GIF | | GOLD_vs_IDEA | | GOLD_vs_MEGU | |
|---|---|---|---|---|---|---|
| Unlearn ratio | 0.1 | 0.8 | 0.1 | 0.8 | 0.1 | 0.8 |
| Cora | 1.27 | 681.97 | 1.27 | 681.97 | 1.27 | 681.38 |
| Citeseer | 1.07 | 439.44 | 1.07 | 439.44 | 1.07 | 439.36 |
| Photo | 1.39 | 307.67 | 1.39 | 307.67 | 1.39 | 307.66 |
| Amazon-Ratings | 1.39 | 500.69 | 1.39 | 500.69 | 1.39 | 500.72 |
| Roman-Empire | 1.40 | 663.72 | 1.40 | 663.72 | 1.40 | 662.94 |
| Ogbn-Arxiv | 1.44 | 429.48 | 1.44 | 429.49 | 1.44 | 428.84 |

## 5.3 RQ2. EFFICIENCY

**Time:** Table 7 reports the running time of each technique. A notable finding is that MEGU, despite being the strongest performer in terms of fidelity, is slower than GOLD on large datasets, and hence, making it impractical as an unlearning algorithm. GRAPHERASER and GUIDE are by far the slowest, as their shard-based preprocessing dominates the pipeline. By contrast, GIF and IDEA are generally faster than GOLD on most datasets. Worryingly, on Reddit, all algorithms except COGNAC and ETR fail to scale due to GPU memory saturation. ETR is the only method to outperform Gold on Reddit.

**The hidden cost of PROJECTOR's Speed:** We observe an interesting trend with PROJECTOR: it is by far the fastest unlearning algorithm. Projector scales polynomially with feature dimensions,

Table 7: Total unlearning time = unlearning time (Table 22) + preprocessing time (Table 23) in seconds. Values in parentheses indicate the percentage of time spent in preprocessing. Cells highlighted in red exceed Gold's runtime. Cognac and Projector does not perform any pre-processing.

| Dataset | GOLD | MEGU | GIF | IDEA | GNNDELETE | GRAPHERASER | GUIDE | PROJECTOR | COGNAC | ETR |
|---|---|---|---|---|---|---|---|---|---|---|
| CORA | 0.65 | 0.61 (100%) | 0.39 (31%) | 0.38 (45%) | 0.60 (23%) | 22.93 (46%) | 77.66 (46%) | 1.69 | 2.16 | 1.65 (45%) |
| CITESEER | 0.66 | 0.63 (98%) | 0.45 (18%) | 0.48 (52%) | 0.62 (23%) | 23.20 (44%) | 81.05 (45%) | 10.57 | 1.00 | 2.51 (30%) |
| PHOTO | 0.90 | 0.51 (100%) | 0.37 (78%) | 0.40 (92%) | 0.87 (28%) | 30.02 (48%) | 919.00 (73%) | 0.47 | 1.45 | 2.02 (44%) |
| AMAZON-R | 1.70 | 2.50 (100%) | 0.74 (54%) | 0.47 (94%) | 1.24 (35%) | 28.70 (44%) | 782.20 (68%) | 0.19 | 3.53 | 1.84 (49%) |
| ROMAN-E | 1.77 | 2.28 (100%) | 0.63 (56%) | 0.54 (46%) | 1.08 (32%) | 27.13 (50%) | 442.20 (75%) | 0.32 | 1.23 | 2.68 (28%) |
| ARXIV | 9.90 | 20.49 (100%) | 4.79 (60%) | 5.31 (68%) | OOM | 116.00 (53%) | OOM | 0.30 | 30.48 | 10.81 (19%) |
| Reddit | 96.73 | OOM | OOM | OOM | OOM | OOT | OOM | OOM | 197.43 | 54.86 (53%) |

Table 8: Peak Allocated Memory (MB) in GPU.

| Dataset | Gold | MEGU | GIF | IDEA | GNNDELETE | GRAPHERASER | PROJECTOR | GUIDE | COGNAC | ETR |
|---|---|---|---|---|---|---|---|---|---|---|
| CORA | 39.11 | 162.85 | 88.67 | 85.78 | 60.05 | 98.08 | 1216.81 | 126.89 | 73.12 | 106 |
| CITESEER | 70.16 | 482.54 | 158.58 | 155.12 | 92.11 | 253.69 | 1666.57 | 291.43 | 172.38 | 196 |
| PHOTO | 151.31 | 251.71 | 551.46 | 543.30 | 756.48 | 317.02 | 4365.23 | 815.00 | 233.41 | 456 |
| AMAZON-RATINGS | 146.74 | 309.37 | 584.48 | 551.41 | 1813.03 | 292.60 | 339.96 | 459.76 | 251.98 | 346 |
| ROMAN-EMPIRE | 92.24 | 278.99 | 407.87 | 405.20 | 680.50 | 205.38 | 100.34 | 259.70 | 172.11 | 284 |
| ogbn-ARXIV | 1290.09 | 1753.91 | 7940.49 | 8263.62 | OOM | 2125.08 | 765.58 | OOM | 1907.15 | 2406 |
| REDDIT | 2861 | OOM | OOM | OOM | OOM | OOT | OOM | OOM | 14464 | 1539 |

which is the dominant factor when feature dimension is larger than the number of nodes in the unlearn request (Cong and Mahdavi, 2023). Hence, it takes more time on cora and citeseer than ogbn-arxiv, a much larger graph. However, this advantage comes with a critical caveat; PROJECTOR operates on a customized linear GNN that is, on average, 100× slower to train than a standard GCN (see Table 15 in Appendix). As a result, the base model itself becomes challenging to train on large datasets. Since the GNN was custom-made to make unlearning fast and exact, reporting unlearning time in isolation obscures these underlying limitations.

**Memory:** Table 8 reports peak GPU memory usage. We find that all unlearning techniques except ETR consume more memory than GOLD. Only ETR and COGNAC are able to scale on Reddit since they support batching. Although batching is not fundamentally impossible on the remaining techniques, current implementations rely on design choices that make batching non-trivial, often leading to memory blowups on large datasets. For instance, MEGU assumes an operator gate that consumes the entire graph and model outputs, making it difficult to redesign the training schema without incurring major changes in runtime and performance. GIF and IDEA rely on full gradient and Hessian-vector products; while approximate batching is theoretically possible, it would trade off accuracy and efficiency, and current implementations provide no stable alternatives. In GNNDELETE, its *deleted-edge consistency* condition, implemented by embedding all edges in the graph, breaks under batching and reindexing, making a tailored batching scheme essential. Finally, GRAPHERASER avoids batching by design through sharding, but its preprocessing overhead is so large that sharding itself dominates runtime, often exceeding the cost of retraining from scratch.

**Runtime Measurement Gaps:** We also observe a trend in existing literature to underestimate the true cost of unlearning. Specifically, when an unlearning request arrives, the total response time is not limited to updating model parameters on the residual graph. It also includes the preprocessing needed to construct the residual graph itself and to extract or organize the information required for the unlearning procedure. This preprocessing pipeline differs substantially between retraining and unlearning algorithms, and prior work has largely ignored these costs in reported runtimes. In MEGU, the reported times exclude the conversion of adjacency matrices to GPU tensors and the selection of $k$-hop neighborhoods needed before optimization. GIF and IDEA similarly account only for influence-function updates, omitting the expensive steps of neighborhood identification and gradient retrieval. GNNDELETE ignores expensive preprocessing steps, including deleting edge masks, forming subgraphs, loss computation, backpropagation, and evaluations. GRAPHERASER, GUIDE separately reports partitioning, unlearning, and aggregation times, but in practice unlearning time for them should be defined as data removal from shards followed by subsequent retraining.

## 5.4 RQ3. FORGETTING

We use three complementary attack families: membership inference, inversion, and noisy-labeler attacks. Together, these attacks probe different facets of residual information, ranging from prediction-level leakage (MIA) to reconstruction of hidden representations (inversion) and label-dependent vulnerability (noisy-labeler attacks).

We first observe from the MIA results (Fig. 1) that most techniques achieve AUROC values close to 0.5, indicating that the adversary cannot reliably distinguish deleted from retained nodes. However,

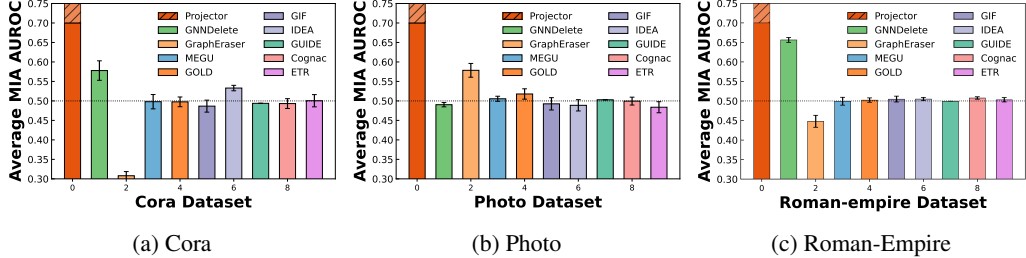

(a) Cora  (b) Photo  (c) Roman-Empire

Figure 1: MIA Histograms. The results on the remaining datasets are available in Fig. 3.

Table 9: AUCROC on Inversion attack (Zhang et al., 2026). An AUCROC close to 0.5 is better.

| Method | CORA | CITESEER | PHOTO | AMAZON-R. | ROMAN-E. | OGBN-ARXIV |
|---|---|---|---|---|---|---|
| MEGU | $0.53 \pm 0.01$ | $0.54 \pm 0.00$ | $0.53 \pm 0.01$ | $0.51 \pm 0.00$ | $0.51 \pm 0.00$ | $0.50 \pm 0.00$ |
| GIF | $0.53 \pm 0.02$ | $0.54 \pm 0.01$ | $0.53 \pm 0.01$ | $0.51 \pm 0.00$ | $0.51 \pm 0.00$ | $0.50 \pm 0.00$ |
| IDEA | $0.52 \pm 0.01$ | $0.54 \pm 0.01$ | $0.53 \pm 0.01$ | $0.51 \pm 0.00$ | $0.51 \pm 0.00$ | $0.50 \pm 0.00$ |
| GNNDELETE | $0.78 \pm 0.05$ | $0.90 \pm 0.01$ | $0.89 \pm 0.12$ | $0.95 \pm 0.01$ | $0.80 \pm 0.01$ | OOM |
| PROJECTOR | $0.60 \pm 0.02$ | $0.57 \pm 0.00$ | $0.53 \pm 0.01$ | $0.60 \pm 0.01$ | $0.52 \pm 0.01$ | $0.56 \pm 0.00$ |
| GRAPHERASER | $0.54 \pm 0.01$ | $0.54 \pm 0.01$ | $0.53 \pm 0.01$ | $0.51 \pm 0.00$ | $0.51 \pm 0.01$ | $0.51 \pm 0.00$ |
| GUIDE | $0.53 \pm 0.00$ | $0.54 \pm 0.01$ | $0.53 \pm 0.01$ | $0.51 \pm 0.00$ | $0.51 \pm 0.00$ | OOM |
| COGNAC | $0.54 \pm 0.00$ | $0.58 \pm 0.01$ | $0.54 \pm 0.01$ | $0.51 \pm 0.00$ | $0.52 \pm 0.00$ | $0.50 \pm 0.00$ |
| ETR | $0.54 \pm 0.00$ | $0.53 \pm 0.01$ | $0.53 \pm 0.01$ | $0.51 \pm 0.00$ | $0.51 \pm 0.00$ | $0.50 \pm 0.00$ |

PROJECTOR, GRAPHERASER, and GNNDELETE deviate noticeably from 0.5, revealing that these methods continue to leak membership information despite unlearning.

The inversion attack results reported in Table 9 sharpen this conclusion. Methods such as MEGU, GIF, IDEA, GRAPHERASER, and GUIDE achieve AUROC values extremely close to 0.5 across all datasets, suggesting that even an adversary with access to logits cannot reconstruct discriminative representations of the deleted nodes. In stark contrast, GNNDELETE exhibits severe vulnerability, with AUROC values ranging from 0.78 to 0.95, and PROJECTOR shows moderate but consistent degradation, further confirming that these approaches leave behind strong representational signatures of the forgotten nodes.

A similar pattern appears under the noisy-labeler attack (Table 24). Nearly all methods maintain AUROC values indistinguishable from random guessing, but GNNDELETE again shows clear susceptibility, particularly on larger graphs such as Amazon-Ratings. The consistency of these failures across all three attack families indicates that certain structural characteristics in GNNDELETE (and, to a lesser extent, PROJECTOR) prevent complete removal of the influence of deleted nodes.

## 5.5 RQ4: ROBUSTNESS TO LARGE-SCALE UNLEARNING

Fig. 2 investigates **RQ4** by examining how accuracy, fidelity and logit distance evolve with increasing unlearn ratios on the CORA dataset. In this setting, curves that remain close to the GOLD retrained model would indicate faithful alignment between unlearning and retraining. Note that nodes are deleted only from the training set. Interestingly, gold accuracy drops to less than 0.2 when 80% of the nodes are removed but MEGU, GIF, and IDEA are largely unaffected. Additionally, we notice that the fidelity of these techniques drops, and logit distance continuously increases with increasing unlearning ratio. Stability of accuracy and continuous degradation of fidelity and logit distance suggests the possibility of unfaithful forgetting under high deletion workloads.

## 5.6 RQ5: ROBUSTNESS TO WORKLOAD DISTRIBUTIONS

**Node deletion:** Table 10 reports fidelity under five strategies: (i) random nodes, (ii) nodes from the most frequent label class, (iii) nodes with second most frequent label class, (iv) low-degree nodes, and (v) high-degree nodes. Two key patterns emerge. Random deletions and those from the highest frequency classes are generally easier to handle. Random deletions preserve

Table 10: Fidelity of different techniques under various node deletion strategies for the Cora dataset (Unlearn ratio 0.1). Higher fidelity indicates closer alignment with GOLD.

| Technique | Random | High Freq | 2nd High Freq | Low Degree | High Degree |
|---|---|---|---|---|---|
| ETR | $0.88 \pm 0.01$ | 0.89 | 0.87 | 0.88 | 0.88 |
| COGNAC | $0.90 \pm 0.01$ | 0.91 | 0.82 | 0.88 | 0.86 |
| MEGU | $0.95 \pm 0.00$ | 0.95 | 0.92 | 0.94 | 0.96 |
| GnnDelete | $0.79 \pm 0.06$ | 0.92 | 0.82 | 0.91 | 0.69 |
| GIF | $0.93 \pm 0.01$ | 0.93 | 0.89 | 0.92 | 0.89 |
| IDEA | $0.92 \pm 0.00$ | 0.91 | 0.88 | 0.90 | 0.89 |
| PROJECTOR | $0.99 \pm 0.00$ | 0.97 | 0.92 | 0.95 | 0.90 |
| GRAPH ERASER | $0.88 \pm 0.00$ | 0.90 | 0.74 | 0.91 | 0.87 |

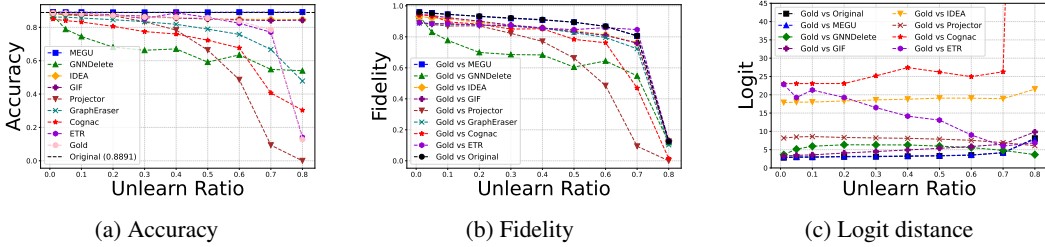

(a) Accuracy        (b) Fidelity        (c) Logit distance

Figure 2: Accuracy vs. Unlearn Ratio plot over all techniques.

the overall topology and label distribution. Deletions from the highest-frequency class have limited impact, e.g., in Cora, this class is twice as large as the second-highest, so its dominance persists even after removing 10% of its nodes. Deletions from low-degree classes, also dont cause much effect because they affect embeddings of very few nodes. In contrast, deleting high-degree nodes destabilizes message passing by removing structural hubs, producing widespread embedding changes. Similarly, removing nodes from the second-most frequent class is disruptive: these nodes are numerous enough to influence learning but not dominant enough to absorb the effect, so their removal shifts class distributions unpredictably. Overall, MEGU demonstrates the highest robustness to deletion distributions. These results also highlight that structured deletion requests reveal vulnerabilities hidden by random removals, highlighting the need to evaluate unlearning under targeted deletion scenarios.

**GNN architecture:** Tables 12-14 in the Appendix evaluate robustness across different GNN backbones. MEGU delivers the most reliable performance across architectures. Overall, we do not observe a clear trend suggesting that any specific architecture is inherently harder to unlearn.

# 6 CONCLUDING INSIGHTS

Table 11 distills the key insights across the three pillars of efficiency and sclability, utility preservation, and faithful forgetting.

**Efficiency and Scalability.** Unlearning is most valuable when retraining is prohibitively expensive. Yet, all algorithms except COGNAC and ETR, fail precisely in this regime (such as REDDIT). Several methods incur substantial preprocessing overheads or rely on computationally intensive subgraph extraction and optimization steps, making them slower than retraining. While MEGU, GIF, and IDEA offer faster unlearning times than retraining, their large memory footprints prohibit them from scaling to Reddit.

**Utility Preservation.** The utility of an unlearned model is ideally measured relative to the retrain-from-scratch "Gold" model. While accuracy remains the most commonly reported proxy, our expanded metrics, fidelity and logit distance, reveal deeper trends. Several methods, such as COGNAC and ETR, achieve competitive accuracy but deviate substantially in logit-space or parameter-space, indicating that the underlying models differ from the Gold solution even when predictions appear similar. In contrast, methods such as MEGU, GIF, and IDEA exhibit consistently high fidelity and low logit divergence.

**Forgetting Quality.** Robustness to attacks is handled well by most methods except PROJECTOR, GNNDELETE, and GRAPHERASER. COGNAC shows hints of topological information leakage in inversion attacks.

**Decision-Making Guidance.** The ideal method is one that has a ✓ on each of the pillars in Table 11. However, no such method exists currently. On small and medium graphs, MEGU, GIF, and IDEA remain the most dependable choices, providing strong accuracy, alignment with GOLD, and robust forgetting. On large graphs where retraining is expensive, ETR currently offers the best compromise: it scales, surpasses retraining in efficiency on the largest dataset, and exhibits solid forgetting performance, albeit with reduced geometric fidelity.

Table 11: Evaluation of graph unlearning techniques against the three pillars: Efficiency, Utility, and Forgetting. A ✓ indicates the pillar is satisfied; a × indicates a violation. ○ indicates partial satisfaction.

| Technique | Efficiency | Utility | Forgetting |
|---|---|---|---|
| GUIDE | × | ✓ | ✓ |
| PROJECTOR | ✓ | ✓ | × |
| GNNDELETE | ✓ | × | × |
| GNNERASER | × | ✓ | × |
| GIF | ○ | ✓ | ✓ |
| IDEA | ○ | ✓ | ✓ |
| MEGU | ○ | ✓ | ✓ |
| COGNAC | × | ○ | ○ |
| ETR | ○ | ○ | ✓ |

ACKNOWLEDGEMENTS

We acknowledge the Yardi School of AI, IIT Delhi for supporting this research. This work was partially supported by the CSE Research Acceleration Fund of IIT Delhi. Samyak Jain acknowledges CSE dept IIT Delhi for sponsoring his travel to ICLR26. Sainyam Galhotra's work was supported by ARO grant W911NF-25-1-0254, BSF grant 2024101, and a grant from Infosys. Any opinions, findings, and conclusions or recommendations expressed in this material are those of the authors and do not necessarily reflect those of the sponsors.

REPRODUCIBILITY

To ensure full reproducibility of our experiments, we provide the following:

**Code and Documentation.**  Our complete codebase is publicly available through a GitHub repository `https://github.com/idea-iitd/Unlearning_Benchmark` . A detailed `README` file is included, which documents installation, usage instructions, and experiment configurations. Each technique is integrated from its publicly released implementation, with general configurations and hyperparameters made accessible through command-line arguments. Default values for all hyperparameters are included in the code, and all scripts are fully commented and seeded for consistent reproducibility.

**Datasets.**  We conduct experiments on widely used benchmark datasets: **Cora**, **Citeseer**, **OGBN-Arxiv**, **Photo**, **Amazon-Ratings**, **Roman-Empire**, and **Reddit** (see Table 2). All datasets are publicly available through standard repositories such as PyTorch Geometric and OGB. Dataset statistics are provided in the main paper for clarity.

**Hyperparameters.**  All hyperparameters (e.g., learning rate, weight decay, number of epochs, unlearning budgets) are specified in the codebase via an argument parser. For each baseline technique, we follow the configurations recommended in the respective papers, ensuring fairness and consistency.

**Environment.**  We provide environment configuration files (e.g., `requirements.txt` and conda `yaml`) that specify all dependencies. Our experiments were run on the following system setup:

- **CPU:** 96 logical cores
- **RAM:** 512 GB
- **GPU:** NVIDIA A100-PCIe-40GB
- **Operating System:** Ubuntu 20.04.4 LTS
- **PyTorch Version:** 1.13.1+cu117
- **CUDA Version:** 11.7
- **PyTorch Geometric Version:** 2.3.1

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

# A  APPENDIX

**Lemma A.1** (Similarity of Activations Under Similar Parameters). *Let $f_\Theta$ be a message-passing GNN with parameters $\Theta$, using standard aggregation functions (mean/sum/max) and activation functions (RELU, sigmoid, or tanh). If $\Theta'$ denotes the parameters of the gold model and $\widetilde{\Theta}$ the parameters of an unlearned model, then the intermediate activations satisfy*

$$\big\|\mathbf{h}^{(\ell)}(\Theta') - \mathbf{h}^{(\ell)}(\widetilde{\Theta})\big\| \ \le \ L_\ell \, \|\Theta' - \widetilde{\Theta}\|,$$

*for some layer-dependent Lipschitz constant $L_\ell > 0$. Thus, if the parameters are close, the activations at every layer are also close.*

*Proof.* Rauchwerger et al. (2025) show that message-passing GNN layers are Lipschitz continuous. Let $L_\ell$ denote the Lipschitz constant of layer $\ell$. For a generic MPNN layer of the form

$$\mathbf{h}_v^{(\ell+1)} = \phi\Big(\mathbf{W}_\ell \cdot \mathrm{AGG}\big(\{\mathbf{h}_u^{(\ell)} : u \in \mathcal{N}(v)\}\big)\Big),$$

the Lipschitz constant satisfies

$$L_\ell \ = \ L_{\mathrm{AGG}} \, \|\mathbf{W}_\ell\|_2,$$

where $L_{\mathrm{AGG}}$ is the Lipschitz constant of the aggregator ($L_{\mathrm{AGG}} = 1$ for mean/max, and $L_{\mathrm{AGG}} \le d_{\max}$ for sum aggregation). Since common activations such as ReLU, $\tanh$, and sigmoid are at most 1-Lipschitz, they do not enlarge the bound.

Because the composition of Lipschitz functions is Lipschitz, the entire $L$-layer GNN satisfies

$$\big\|\mathbf{h}^{(L)}(\Theta') - \mathbf{h}^{(L)}(\widetilde{\Theta})\big\| \ \le \ \left(\prod_{\ell=0}^{L-1} L_\ell\right) \|\Theta' - \widetilde{\Theta}\|.$$

Thus, if the parameters of the Gold model $\Theta'$ and the unlearned model $\widetilde{\Theta}$ are close, then their intermediate activations at every layer are also guaranteed to be close, up to the (typically small) product of per-layer Lipschitz constants.[1]  □

## A.1  EXPERIMENTAL SETUP

For the baseline algorithms, we use the code available at *OpenGU* Fan et al. (2025). Our codebase and an integrated framework to evaluate all baselines is released at `https://github.com/idea-iitd/Unlearning_Benchmark`

### A.1.1  HARDWARE CONFIGURATION

All experiments were conducted on a high-performance computing system with the following specifications:

- **CPU:** 96 logical cores
- **RAM:** 512 GB
- **GPU:** NVIDIA A100-PCIE-40GB

### A.1.2  SOFTWARE CONFIGURATION

The software environment for our experiments was configured as follows:

- **Operating System:** Linux (Ubuntu 20.04.4 LTS (GNU/Linux 5.4.0-124-generic x86_64))
- **PyTorch Version:** 1.13.1+cu117
- **CUDA Version:** 11.7
- **PyTorch Geometric Version:** 2.3.1

---

[1] Modern GNNs are shallow, with $L \le 3$, so the Lipschitz product does not explode in practice.

Table 12: Test accuracies (± std) of different unlearning methods across datasets and base models. "OOM" denotes out-of-memory. Best values are highlighted among unlearning methods only.

| Dataset | Base Model | Original | GOLD | MEGU | GNN Delete | GIF | IDEA | GraphEraser | GST |
|---|---|---|---|---|---|---|---|---|---|
| Cora | GCN | 0.89 ± 0.00 | 0.88 ± 0.00 | 0.89 ± 0.00 | 0.76 ± 0.05 | 0.88 ± 0.00 | 0.87 ± 0.00 | 0.85 ± 0.00 | OOM |
| | GAT | 0.89 ± 0.00 | 0.88 ± 0.00 | 0.88 ± 0.01 | 0.78 ± 0.02 | 0.84 ± 0.01 | 0.84 ± 0.01 | 0.85 ± 0.01 | OOM |
| | GIN | 0.88 ± 0.00 | 0.88 ± 0.01 | 0.87 ± 0.00 | 0.71 ± 0.05 | 0.82 ± 0.01 | 0.82 ± 0.01 | 0.83 ± 0.00 | OOM |
| Citeseer | GCN | 0.77 ± 0.00 | 0.73 ± 0.01 | 0.77 ± 0.00 | 0.76 ± 0.01 | 0.76 ± 0.00 | 0.77 ± 0.00 | 0.77 ± 0.01 | OOM |
| | GAT | 0.75 ± 0.00 | 0.74 ± 0.01 | 0.74 ± 0.01 | 0.72 ± 0.01 | 0.72 ± 0.01 | 0.72 ± 0.01 | 0.73 ± 0.01 | OOM |
| | GIN | 0.76 ± 0.00 | 0.74 ± 0.01 | 0.76 ± 0.01 | 0.71 ± 0.02 | 0.71 ± 0.01 | 0.71 ± 0.01 | 0.75 ± 0.00 | OOM |
| Photo | GCN | 0.92 ± 0.00 | 0.93 ± 0.00 | 0.92 ± 0.00 | 0.34 ± 0.11 | 0.92 ± 0.00 | 0.93 ± 0.00 | 0.92 ± 0.00 | OOM |
| | GAT | 0.93 ± 0.00 | 0.83 ± 0.09 | 0.90 ± 0.02 | 0.37 ± 0.14 | 0.90 ± 0.00 | 0.89 ± 0.00 | 0.73 ± 0.06 | OOM |
| | GIN | 0.78 ± 0.00 | 0.76 ± 0.08 | 0.78 ± 0.00 | 0.14 ± 0.05 | 0.75 ± 0.00 | 0.76 ± 0.00 | 0.89 ± 0.00 | OOM |
| Amazon-R. | GCN | 0.43 ± 0.00 | 0.43 ± 0.01 | 0.42 ± 0.00 | 0.37 ± 0.00 | 0.43 ± 0.00 | 0.41 ± 0.00 | 0.44 ± 0.00 | OOM |
| | GAT | 0.46 ± 0.00 | 0.46 ± 0.00 | 0.45 ± 0.01 | 0.38 ± 0.00 | 0.40 ± 0.01 | 0.41 ± 0.01 | 0.41 ± 0.00 | OOM |
| | GIN | 0.45 ± 0.00 | 0.43 ± 0.00 | 0.42 ± 0.00 | 0.38 ± 0.01 | 0.43 ± 0.00 | 0.43 ± 0.00 | 0.41 ± 0.01 | OOM |
| Roman-E. | GCN | 0.41 ± 0.00 | 0.41 ± 0.00 | 0.41 ± 0.00 | 0.33 ± 0.00 | 0.44 ± 0.00 | 0.48 ± 0.00 | 0.41 ± 0.00 | OOM |
| | GAT | 0.52 ± 0.00 | 0.49 ± 0.01 | 0.41 ± 0.02 | 0.34 ± 0.01 | 0.41 ± 0.01 | 0.41 ± 0.00 | 0.28 ± 0.00 | OOM |
| | GIN | 0.45 ± 0.00 | 0.44 ± 0.00 | 0.45 ± 0.00 | 0.33 ± 0.00 | 0.43 ± 0.00 | 0.43 ± 0.00 | 0.36 ± 0.00 | OOM |

### A.1.3 HYPERPARAMETERS

For all the algorithms, we have used their default unlearning arguments and hyperparameters. We use the same baseline GCN for all techniques and consistent seeds to across techniques to reduce bias and ensure fairness. More details of the parameters for each algo can be seen in our codebase.

## A.2 FORGETTING VIA MEMBERSHIP INFERENCE ATTACKS

MIAs test whether an unlearned model can still leak information about the deleted nodes' training membership. In the standard formulation, an attack model is trained to distinguish members (training samples) from non-members (test samples) using the posterior distributions (class probabilities) of a target model as input.

We adapt this methodology to the unlearning setting by designing a shadow attack model based on the *change* in posteriors before and after unlearning. Specifically, for each node we compute the $\ell_2$ norm between its posterior probability vector produced by the original model and the corresponding vector after unlearning. Intuitively, member nodes (those originally in the training set and later targeted for unlearning) should undergo larger shifts in their posteriors than non-member nodes. We then label deleted training nodes as positive instances (members) and an equal number of test nodes as negative instances (non-members). Using these labels and the posterior-change scores, we compute the AUROC. AUROC quantifies the attack's discriminative power: values close to 0.5 indicate random guessing and thus strong protection, while higher values imply that the unlearned model still leaks membership information and has not fully forgotten.

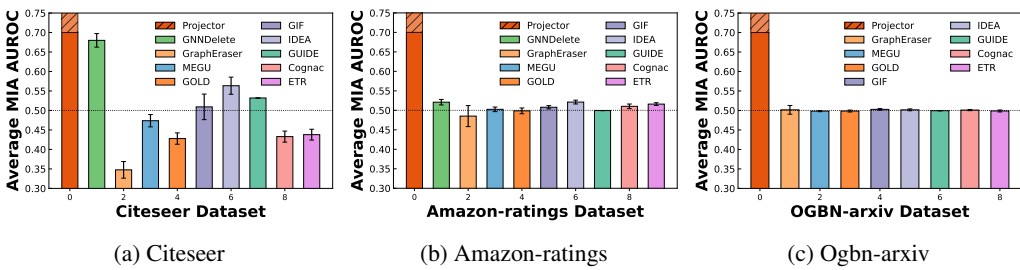

(a) Citeseer      (b) Amazon-ratings      (c) Ogbn-arxiv

Figure 3: MIA Histograms to evaluate forgetting.

Table 13: Fidelity results across datasets, base models, and unlearning methods. PROJECTOR is ommitted since it cannot handle generic GNNs.

| Dataset | Base Model | MEGU | GIF | IDEA | GNN Delete | GraphEraser | GST |
|---------|-----------|------|-----|------|------------|-------------|-----|
| Cora | GCN | 0.95 ± 0.00 | 0.93 ± 0.01 | 0.92 ± 0.00 | 0.79 ± 0.06 | 0.88 ± 0.00 | OOM |
| | GAT | 0.93 ± 0.01 | 0.87 ± 0.01 | 0.87 ± 0.00 | 0.81 ± 0.02 | 0.90 ± 0.01 | OOM |
| | GIN | 0.95 ± 0.01 | 0.86 ± 0.01 | 0.86 ± 0.01 | 0.75 ± 0.06 | 0.89 ± 0.02 | OOM |
| Citeseer | GCN | 0.85 ± 0.01 | 0.82 ± 0.01 | 0.83 ± 0.01 | 0.83 ± 0.01 | 0.81 ± 0.01 | OOM |
| | GAT | 0.89 ± 0.01 | 0.84 ± 0.01 | 0.83 ± 0.01 | 0.88 ± 0.01 | 0.88 ± 0.01 | OOM |
| | GIN | 0.89 ± 0.01 | 0.81 ± 0.01 | 0.81 ± 0.01 | 0.81 ± 0.02 | 0.82 ± 0.01 | OOM |
| Photo | GCN | 0.99 ± 0.00 | 0.97 ± 0.00 | 0.95 ± 0.00 | 0.35 ± 0.12 | 0.98 ± 0.00 | OOM |
| | GAT | 0.84 ± 0.07 | 0.84 ± 0.08 | 0.84 ± 0.08 | 0.40 ± 0.13 | 0.77 ± 0.09 | OOM |
| | GIN | 0.75 ± 0.05 | 0.72 ± 0.05 | 0.72 ± 0.06 | 0.12 ± 0.08 | 0.79 ± 0.10 | OOM |
| Amazon-R. | GCN | 0.84 ± 0.03 | 0.87 ± 0.01 | 0.82 ± 0.01 | 0.67 ± 0.01 | 0.85 ± 0.01 | OOM |
| | GAT | 0.79 ± 0.03 | 0.62 ± 0.01 | 0.64 ± 0.01 | 0.59 ± 0.03 | 0.71 ± 0.03 | OOM |
| | GIN | 0.76 ± 0.04 | 0.72 ± 0.02 | 0.72 ± 0.02 | 0.66 ± 0.04 | 0.77 ± 0.02 | OOM |
| Roman-E. | GCN | 0.86 ± 0.00 | 0.57 ± 0.00 | 0.53 ± 0.00 | 0.62 ± 0.01 | 0.60 ± 0.01 | OOM |
| | GAT | 0.68 ± 0.03 | 0.44 ± 0.00 | 0.44 ± 0.01 | 0.49 ± 0.01 | 0.49 ± 0.01 | OOM |
| | GIN | 0.83 ± 0.01 | 0.53 ± 0.00 | 0.53 ± 0.00 | 0.55 ± 0.01 | 0.65 ± 0.00 | OOM |

Table 14: Logit distance results across datasets, base models, and unlearning methods (lower is better). PROJECTOR and GRAPHERASER are omitted since PROJECTOR does not handle heneric GNNs and GRAPHERASER uses sharding.

| Dataset | Base Model | MEGU | GIF | IDEA | GNN Delete | GST |
|---------|-----------|------|-----|------|------------|-----|
| Cora | GCN | 3.05 ± 0.20 | 3.60 ± 0.19 | 18.03 ± 0.80 | 5.91 ± 0.26 | OOM |
| | GAT | 10.86 ± 2.01 | 10.74 ± 1.49 | 10.68 ± 1.49 | 12.05 ± 1.39 | OOM |
| | GIN | 3.86 ± 0.28 | 6.38 ± 0.23 | 6.38 ± 0.23 | 7.36 ± 0.28 | OOM |
| Citeseer | GCN | 6.31 ± 0.23 | 6.33 ± 0.23 | 10.06 ± 0.44 | 7.36 ± 0.19 | OOM |
| | GAT | 12.74 ± 2.35 | 15.52 ± 3.60 | 15.45 ± 3.65 | 13.10 ± 3.10 | OOM |
| | GIN | 5.32 ± 0.65 | 7.23 ± 0.91 | 7.23 ± 0.91 | 8.07 ± 0.95 | OOM |
| Photo | GCN | 3.81 ± 0.68 | 4.20 ± 0.61 | 22.79 ± 1.88 | 8.57 ± 0.46 | OOM |
| | GAT | 23.90 ± 20.07 | 23.39 ± 20.34 | 23.35 ± 20.34 | 23.32 ± 20.46 | OOM |
| | GIN | 34.75 ± 2.96 | 25.62 ± 3.53 | 25.54 ± 3.61 | 64.22 ± 24.42 | OOM |
| Amazon-R. | GCN | 0.59 ± 0.15 | 0.56 ± 0.12 | 4.69 ± 0.19 | 1.10 ± 0.06 | OOM |
| | GAT | 1.32 ± 0.34 | 1.70 ± 0.07 | 1.69 ± 0.07 | 1.44 ± 0.11 | OOM |
| | GIN | 1.61 ± 0.48 | 1.83 ± 0.36 | 1.84 ± 0.36 | 2.04 ± 0.46 | OOM |
| Roman-E. | GCN | 1.57 ± 0.24 | 5.24 ± 0.13 | 22.62 ± 0.64 | 2.64 ± 0.16 | OOM |
| | GAT | 6.91 ± 0.95 | 19.42 ± 0.60 | 19.36 ± 0.62 | 7.11 ± 0.52 | OOM |
| | GIN | 2.42 ± 0.10 | 5.97 ± 0.09 | 5.97 ± 0.09 | 3.93 ± 0.08 | OOM |

Table 15: Accuracies and training times achieved by the Original model and GOLD model in Projector and vanilla GCN.

| Dataset | Original Acc. | GOLD Acc. | GCN Training Time | Pro_GNN Training Time |
|---------|---------------|-----------|-------------------|------------------------|
| Cora | 0.8392 ± 0.0000 | 0.8362 ± 0.0043 | 0.4822 ± 0.3140 | 33.3136 ± 2.4494 |
| Citeseer | 0.7639 ± 0.0000 | 0.7654 ± 0.0016 | 0.4765 ± 0.3378 | 53.8463 ± 32.8146 |
| Photo | 0.8693 ± 0.0000 | 0.8701 ± 0.0017 | 0.5703 ± 0.3294 | 220.2981 ± 10.2529 |
| Amazon-ratings | 0.4741 ± 0.0000 | 0.4723 ± 0.0023 | 1.0025 ± 0.3326 | 288.1078 ± 405.8160 |
| Roman-empire | 0.4947 ± 0.0000 | 0.4969 ± 0.0023 | 0.9746 ± 0.3257 | 90.0904 ± 0.8290 |
| OGBN-arixv | 0.6126 ± 0.0000 | 0.6124 ± 0.0009 | 10.2016 ± 0.4677 | 11856.62 ± 1018.27 |

## A.3 EDGE UNLEARNING

In Tables 16–18, we report accuracy, fidelity, and logit-level L2 distance relative to the gold standard under the setting where 10% edges are deleted uniformly at random. The results exhibit trends

consistent with the node-deletion setting: MEGU continues to deliver stable and competitive performance, typically achieving the best results or performing very close to the top method.

Table 16: Accuracies for edge-deletion experiment (higher is better). Best results are shaded.

| Technique | CORA | CITESEER | PHOTO | AMAZON-R. | ROMAN-E. |
|---|---|---|---|---|---|
| Original | 0.88 ± 0.00 | 0.76 ± 0.00 | 0.92 ± 0.01 | 0.43 ± 0.00 | 0.41 ± 0.00 |
| GOLD | 0.88 ± 0.00 | 0.73 ± 0.00 | 0.92 ± 0.01 | 0.43 ± 0.01 | 0.41 ± 0.00 |
| MEGU | 0.88 ± 0.00 | 0.76 ± 0.00 | 0.91 ± 0.00 | 0.42 ± 0.00 | 0.40 ± 0.00 |
| GIF | 0.89 ± 0.00 | 0.76 ± 0.02 | 0.92 ± 0.02 | 0.43 ± 0.02 | 0.37 ± 0.00 |
| IDEA | 0.88 ± 0.00 | 0.76 ± 0.01 | 0.92 ± 0.00 | 0.42 ± 0.00 | 0.46 ± 0.00 |
| GNNDELETE | 0.62 ± 0.07 | 0.70 ± 0.03 | 0.26 ± 0.06 | 0.36 ± 0.00 | 0.19 ± 0.00 |
| GRAPHERASER | 0.88 ± 0.00 | 0.76 ± 0.01 | 0.92 ± 0.02 | 0.43 ± 0.00 | 0.41 ± 0.00 |
| GUIDE | 0.83 ± 0.00 | 0.75 ± 0.00 | 0.90 ± 0.00 | 0.42 ± 0.00 | 0.37 ± 0.00 |
| COGNAC | 0.83 ± 0.01 | 0.68 ± 0.01 | 0.91 ± 0.01 | 0.45 ± 0.01 | 0.48 ± 0.01 |

Table 17: Fidelity scores for edge-deletion experiment (higher is better). Best results are shaded.

| Technique | CORA | CITESEER | PHOTO | AMAZON-R. | ROMAN-E. |
|---|---|---|---|---|---|
| MEGU | 0.96 ± 0.01 | 0.86 ± 0.01 | 0.97 ± 0.01 | 0.88 ± 0.02 | 0.82 ± 0.01 |
| GIF | 0.95 ± 0.01 | 0.85 ± 0.01 | 0.98 ± 0.00 | 0.92 ± 0.01 | 0.73 ± 0.00 |
| IDEA | 0.94 ± 0.01 | 0.86 ± 0.01 | 0.96 ± 0.00 | 0.85 ± 0.00 | 0.69 ± 0.01 |
| GNNDELETE | 0.64 ± 0.08 | 0.77 ± 0.03 | 0.27 ± 0.06 | 0.64 ± 0.01 | 0.30 ± 0.00 |
| GRAPHERASER | 0.95 ± 0.01 | 0.84 ± 0.01 | 0.99 ± 0.00 | 0.92 ± 0.00 | 0.87 ± 0.01 |
| GUIDE | 0.84 ± 0.01 | 0.79 ± 0.00 | 0.96 ± 0.00 | 0.81 ± 0.00 | 0.69 ± 0.02 |
| COGNAC | 0.92 ± 0.00 | 0.87 ± 0.00 | 0.97 ± 0.01 | 0.75 ± 0.004 | 0.64 ± 0.004 |

Table 18: L2 distance for edge-deletion experiment (lower is better). Best values are shaded.

| Technique | CORA | CITESEER | PHOTO | AMAZON-R. | ROMAN-E. |
|---|---|---|---|---|---|
| MEGU | 2.89 ± 0.20 | 6.23 ± 0.21 | 4.64 ± 0.81 | 0.53 ± 0.07 | 1.51 ± 0.15 |
| GIF | 2.94 ± 0.19 | 6.21 ± 0.22 | 3.86 ± 0.67 | 0.36 ± 0.10 | 2.17 ± 0.13 |
| IDEA | 17.88 ± 0.83 | 10.01 ± 0.51 | 21.99 ± 0.94 | 4.65 ± 0.19 | 17.82 ± 0.62 |
| GNNDELETE | 7.08 ± 0.18 | 7.84 ± 0.16 | 8.98 ± 0.33 | 1.04 ± 0.02 | 3.35 ± 0.07 |
| COGNAC | 16.35 ± 1.02 | 21.86 ± 0.07 | 6.08 ± 0.41 | 0.75 ± 0.005 | 5.74 ± 0.00 |

## A.4 FEATURE UNLEARNING

Following the standard feature-unlearning protocol Li et al. (2024), we sample $10\%$ of the nodes uniformly at random and mask out their features. The corresponding results are reported in Tables 19–21. The overall patterns remain consistent with earlier observations: MEGU continues to deliver the most robust performance across all metrics.

## A.5 PERFORMANCE OF LINEAR GNNS

**Certified Unlearning.** We evaluate the two certified graph-unlearning algorithms, CGU and SCALEGUN, using the official implementations released by their authors. Both methods offer *formal guarantees of exact forgetting*, but these guarantees hold only under highly restrictive assumptions: (i) the underlying GNN must be a custom linear message-passing architecture, and (ii) the prediction task must be *binary* node classification. As a result, neither method supports multi-class node classification natively. For SCALEGUN, multi-class datasets must be decomposed into a one-vs-rest collection of binary tasks; CGU imposes similar constraints due to the structure of its certified update rule. Consequently, the metrics produced by these algorithms are *not directly comparable* to those of unlearning methods that operate in the full multi-class setting.

Table 19: Accuracies for the feature-deletion experiment (higher is better). Best results are shaded.

| Technique | CORA | CITESEER | PHOTO | AMAZON-R. | ROMAN-E. |
|---|---|---|---|---|---|
| Original | $0.89 \pm 0.00$ | $0.77 \pm 0.00$ | $0.92 \pm 0.00$ | $0.44 \pm 0.00$ | $0.41 \pm 0.00$ |
| GOLD | $0.89 \pm 0.01$ | $0.73 \pm 0.00$ | $0.92 \pm 0.00$ | $0.43 \pm 0.00$ | $0.40 \pm 0.00$ |
| MEGU | $0.89 \pm 0.00$ | $0.77 \pm 0.00$ | $0.92 \pm 0.00$ | $0.43 \pm 0.01$ | $0.41 \pm 0.00$ |
| GIF | $0.89 \pm 0.00$ | $0.77 \pm 0.02$ | $0.93 \pm 0.00$ | $0.44 \pm 0.00$ | $0.42 \pm 0.00$ |
| IDEA | $0.88 \pm 0.00$ | $0.77 \pm 0.02$ | $0.93 \pm 0.00$ | $0.42 \pm 0.00$ | $0.49 \pm 0.00$ |
| GNNDELETE | $0.77 \pm 0.03$ | $0.76 \pm 0.01$ | $0.35 \pm 0.06$ | $0.37 \pm 0.00$ | $0.31 \pm 0.00$ |
| GRAPHERASER | $0.89 \pm 0.01$ | $0.77 \pm 0.01$ | $0.93 \pm 0.01$ | $0.44 \pm 0.01$ | $0.42 \pm 0.00$ |
| GUIDE | $0.84 \pm 0.01$ | $0.75 \pm 0.01$ | $0.91 \pm 0.00$ | $0.42 \pm 0.00$ | $0.38 \pm 0.01$ |

Table 20: Fidelity scores for feature-deletion experiment (higher is better). Best results are shaded.

| Technique | CORA | CITESEER | PHOTO | AMAZON-R. | ROMAN-E. |
|---|---|---|---|---|---|
| MEGU | $0.96 \pm 0.00$ | $0.85 \pm 0.00$ | $0.98 \pm 0.00$ | $0.87 \pm 0.03$ | $0.86 \pm 0.00$ |
| GIF | $0.95 \pm 0.01$ | $0.85 \pm 0.01$ | $0.98 \pm 0.01$ | $0.89 \pm 0.01$ | $0.80 \pm 0.01$ |
| IDEA | $0.94 \pm 0.00$ | $0.85 \pm 0.00$ | $0.95 \pm 0.00$ | $0.83 \pm 0.01$ | $0.71 \pm 0.01$ |
| GNNDELETE | $0.81 \pm 0.03$ | $0.83 \pm 0.02$ | $0.36 \pm 0.06$ | $0.68 \pm 0.01$ | $0.55 \pm 0.02$ |
| GRAPHERASER | $0.95 \pm 0.01$ | $0.85 \pm 0.00$ | $0.98 \pm 0.02$ | $0.89 \pm 0.01$ | $0.80 \pm 0.01$ |
| GUIDE | $0.84 \pm 0.01$ | $0.79 \pm 0.00$ | $0.96 \pm 0.00$ | $0.82 \pm 0.01$ | $0.69 \pm 0.02$ |

Empirically, both CGU and SCALEGUN exhibit competitive accuracy and high fidelity on homophilous datasets such as CORA, CITESEER, PHOTO, and OGBN-ARXIV, with their unlearned performance remaining close to the Gold (retrain-from-scratch) model. However, their behavior diverges sharply on heterophilous datasets. SCALEGUN suffers substantial drops in accuracy and fidelity on AMAZON-RATINGS and ROMAN-EMPIRE, indicating brittle performance outside of the linear, homophilous regime for which it is designed. In contrast, CGU remains comparatively stable across these datasets, showing no analogous collapse in utility.

Overall, our evaluation highlights a common limitation of certified graph-unlearning algorithms: although theoretically appealing, they currently apply only to narrow classes of linear binary GNNs and do not yet offer a practical, general-purpose solution for real-world multi-class graph learning. Their results should therefore be interpreted as establishing a theoretical baseline, rather than as direct competitors to unlearning methods that operate on expressive, modern GNN architectures.

## A.6 DISCLAIMER

We note that LLM was used to polish the writing. The LLM output was not used as is but was proofread for both grammatical and semantic correctness by the authors before being used in our manuscript.

Table 21: L2 distance for feature-deletion experiment (lower is better). Best results are shaded.

| Technique | CORA | CITESEER | PHOTO | AMAZON-R. | ROMAN-E. |
|---|---|---|---|---|---|
| MEGU | $3.49 \pm 0.29$ | $7.16 \pm 0.23$ | $4.62 \pm 0.99$ | $0.48 \pm 0.05$ | $1.57 \pm 0.24$ |
| GIF | $3.81 \pm 0.25$ | $7.39 \pm 0.22$ | $4.89 \pm 0.92$ | $0.47 \pm 0.07$ | $2.03 \pm 0.22$ |
| IDEA | $16.85 \pm 0.85$ | $10.49 \pm 0.48$ | $20.89 \pm 2.34$ | $4.60 \pm 0.22$ | $17.22 \pm 0.68$ |
| GNNDELETE | $6.27 \pm 0.27$ | $8.15 \pm 0.19$ | $9.96 \pm 0.38$ | $1.12 \pm 0.06$ | $2.70 \pm 0.18$ |

Table 22: Unlearning times (in seconds) for each unlearning method across datasets. We ignore Cognac and Projector in this table since they do not have a pre-processing step. Their full unlearning times have already been reported in Table 7.

| Dataset | GOLD | MEGU | GIF | IDEA | GNNDELETE | GRAPHERASER | GUIDE | ETR |
|---|---|---|---|---|---|---|---|---|
| CORA | $0.38 \pm 0.02$ | $0.00 \pm 0.00$ | $0.27 \pm 0.01$ | $0.21 \pm 0.04$ | $0.46 \pm 0.01$ | $12.46 \pm 4.16$ | $41.93 \pm 2.65$ | $0.91 \pm 0.08$ |
| CITESEER | $0.39 \pm 0.01$ | $0.01 \pm 0.01$ | $0.37 \pm 0.01$ | $0.23 \pm 0.03$ | $0.48 \pm 0.01$ | $12.93 \pm 4.33$ | $44.36 \pm 0.65$ | $1.76 \pm 0.03$ |
| PHOTO | $0.50 \pm 0.00$ | $0.00 \pm 0.00$ | $0.08 \pm 0.00$ | $0.03 \pm 0.00$ | $0.63 \pm 0.02$ | $15.58 \pm 5.61$ | $252.35 \pm 1.90$ | $1.14 \pm 0.04$ |
| AMAZON-R | $0.73 \pm 0.04$ | $0.00 \pm 0.00$ | $0.34 \pm 0.03$ | $0.03 \pm 0.00$ | $0.81 \pm 0.01$ | $16.07 \pm 4.98$ | $249.74 \pm 3.00$ | $0.94 \pm 0.09$ |
| ROMAN-E | $0.67 \pm 0.03$ | $0.00 \pm 0.00$ | $0.28 \pm 0.02$ | $0.29 \pm 0.01$ | $0.73 \pm 0.07$ | $13.57 \pm 4.33$ | $110.97 \pm 4.35$ | $1.94 \pm 0.08$ |
| ARXIV | $5.01 \pm 0.04$ | $0.00 \pm 0.00$ | $1.91 \pm 0.04$ | $1.71 \pm 0.02$ | OOM | $55.07 \pm 8.86$ | OOM | $8.71 \pm 0.86$ |

Table 23: Unlearning preprocessing times (in seconds) for each unlearning method across datasets. We ignore Cognac and Projector in this table since they do not have a preprocessing step.

| Dataset | GOLD | MEGU | GIF | IDEA | GNNDELETE | GRAPHERASER | GUIDE | ETR |
|---|---|---|---|---|---|---|---|---|
| CORA | $0.27 \pm 0.06$ | $0.61 \pm 0.07$ | $0.12 \pm 0.05$ | $0.17 \pm 0.08$ | $0.14 \pm 0.09$ | $10.47 \pm 4.38$ | $35.73 \pm 4.10$ | $0.74 \pm 0.02$ |
| CITESEER | $0.27 \pm 0.06$ | $0.62 \pm 0.09$ | $0.08 \pm 0.05$ | $0.25 \pm 0.07$ | $0.14 \pm 0.09$ | $10.27 \pm 6.47$ | $36.69 \pm 3.10$ | $0.75 \pm 0.00$ |
| PHOTO | $0.40 \pm 0.09$ | $0.51 \pm 0.09$ | $0.29 \pm 0.07$ | $0.37 \pm 0.07$ | $0.24 \pm 0.06$ | $14.44 \pm 5.15$ | $666.65 \pm 92.60$ | $0.88 \pm 0.00$ |
| AMAZON-R | $0.97 \pm 0.21$ | $2.50 \pm 0.27$ | $0.40 \pm 0.11$ | $0.44 \pm 0.07$ | $0.43 \pm 0.14$ | $12.63 \pm 5.96$ | $532.46 \pm 3.23$ | $0.90 \pm 0.09$ |
| ROMAN-E | $1.10 \pm 0.15$ | $2.28 \pm 0.25$ | $0.35 \pm 0.04$ | $0.25 \pm 0.05$ | $0.35 \pm 0.01$ | $13.56 \pm 7.33$ | $331.23 \pm 25.86$ | $0.74 \pm 0.03$ |
| ARXIV | $4.89 \pm 1.05$ | $20.49 \pm 2.34$ | $2.88 \pm 0.45$ | $3.60 \pm 0.46$ | OOM | $60.93 \pm 14.29$ | OOM | $2.1 \pm 0.27$ |

Table 24: AUROC following Noisy-Labeler attacksSui et al. (2025). (Closer to 0.50 is better.)

| Method | CORA | CITESEER | PHOTO | AMAZON-R. | ROMAN-E. | OGBN-ARXIV |
|---|---|---|---|---|---|---|
| MEGU | $0.50 \pm 0.01$ | $0.51 \pm 0.02$ | $0.50 \pm 0.01$ | $0.50 \pm 0.00$ | $0.50 \pm 0.01$ | $0.50 \pm 0.00$ |
| GIF | $0.50 \pm 0.01$ | $0.51 \pm 0.02$ | $0.50 \pm 0.01$ | $0.51 \pm 0.01$ | $0.50 \pm 0.01$ | $0.50 \pm 0.00$ |
| IDEA | $0.50 \pm 0.01$ | $0.50 \pm 0.02$ | $0.50 \pm 0.01$ | $0.50 \pm 0.01$ | $0.50 \pm 0.01$ | $0.50 \pm 0.00$ |
| GNNDELETE | $0.58 \pm 0.03$ | $0.59 \pm 0.02$ | $0.61 \pm 0.06$ | $0.81 \pm 0.02$ | $0.52 \pm 0.02$ | OOM |
| PROJECTOR | $0.50 \pm 0.03$ | $0.51 \pm 0.04$ | $0.50 \pm 0.01$ | $0.50 \pm 0.00$ | $0.50 \pm 0.01$ | $0.50 \pm 0.00$ |
| GRAPHERASER | $0.49 \pm 0.01$ | $0.52 \pm 0.01$ | $0.50 \pm 0.01$ | $0.50 \pm 0.00$ | $0.50 \pm 0.01$ | $0.50 \pm 0.00$ |
| GUIDE | $0.50 \pm 0.01$ | $0.51 \pm 0.02$ | $0.50 \pm 0.01$ | $0.51 \pm 0.00$ | $0.50 \pm 0.01$ | OOM |
| COGNAC | $0.51 \pm 0.00$ | $0.50 \pm 0.01$ | $0.50 \pm 0.01$ | $0.50 \pm 0.00$ | $0.50 \pm 0.01$ | $0.50 \pm 0.01$ |
| ETR | $0.51 \pm 0.01$ | $0.50 \pm 0.02$ | $0.50 \pm 0.01$ | $0.50 \pm 0.00$ | $0.50 \pm 0.0$ | $0.50 \pm 0.0$ |

Table 25: SCALEGUN unlearning results. Note: SCALEGUN supports only binary classification; for multi-class datasets, it uses one-vs-rest reduction. Hence, results are not directly comparable with other unlearning algorithms.

| Dataset | Orig. Acc | Gold Acc | Unlearned Acc | Fidelity | Logit Dist. |
|---|---|---|---|---|---|
| CORA | 81.7 | 82.2 | 81.5 | 92.6 | 0.51 |
| CITESEER | 75.9 | 75.48 | 74.7 | 93.8 | 0.602 |
| PHOTO | 50.9 | 50.7 | 50.7 | 99.0 | 0.12 |
| AMAZON-RATINGS | 32.2 | 31.7 | 20.6 | 5.8 | 0.03 |
| ROMAN-EMPIRE | 42.7 | 42.6 | 15.3 | 5.6 | 0.54 |
| OGBN-ARXIV | 40.64 | 40.69 | 40.67 | 99.0 | 0.192 |

Table 26: CGU unlearning results. Note: CGU supports only binary classification; for multi-class datasets, it uses one-vs-rest reduction. Hence, results are not directly comparable with other unlearning algorithms.

| Dataset | Orig. Acc | Gold Acc | Unlearnt Acc | Fidelity | Logit Dist. |
|---|---|---|---|---|---|
| CORA | 87.4 | 87.6 | 87.4 | 99.0 | 0.12 |
| CITESEER | 75.9 | 75.9 | 75.1 | 97.1 | 0.144 |
| PHOTOS | 61.9 | 62.2 | 62.2 | 99.8 | 0.20 |
| AMAZON-RATINGS | 35.0 | 34.7 | 35.2 | 96.3 | 0.05 |
| ROMAN-EMPIRE | 28.4 | 28.3 | 28.4 | 97.1 | 0.06 |
| OGBN-ARXIV | 47.8 | 47.9 | 47.9 | 99.9 | 0.02 |

