# OpenReview forum: "Is Graph Unlearning Ready for Practice? A Benchmark on Efficiency, Utility, and Forgetting"
_ICLR.cc/2026/Conference — ICLR 2026 Poster_

### Official Review · Reviewer_MdZs · 2025-10-27

**Soundness:** 3
**Presentation:** 4
**Contribution:** 2
**Rating:** 6
**Confidence:** 4

**Summary:**

This paper presents a benchmark for Graph Neural Network (GNN) unlearning methods, evaluating them based on Efficiency (vs. retraining), Utility (multi-level alignment with retrained models), and Forgetting (MIA resistance). It assesses various existing unlearning techniques across diverse datasets and deletion scenarios. This concludes that most methods suffer from scalability issues (memory, lack of batching), making retraining often preferable for large graphs, and provides guidelines for selecting methods when unlearning is viable. Codebase is also provided.

**Strengths:**

1. **Comprehensive Evaluation:** Establishes a benchmark (efficiency, multi-level utility, forgetting, robustness) that (in my opinion, having worked in this field) advances evaluation standards in GNN unlearning.
2. **Practical Relevance:** Directly compares against retraining, assesses scalability on large graphs, and considers hidden computational costs, providing a reality check on practical viability.
3. **Clear Insights & Guidance:** Systematically reveals limitations (especially scalability/batching) of current methods and provides clear, evidence-based recommendations summarized effectively (Table 10).
4. **Clarity and Reproducibility:** Very well-written and organized, making comparisons accessible. Also provides an open-source codebase implementing multiple methods within its framework.

**Weaknesses:**

1. **Limited Scope of "Forgetting":** Relies solely on one type of MIA based on posterior shifts. Forgetting is a complex concept, and exploring other attack types (ex., attribute inference, link re-inference) or information-theoretic measures could provide a more complete picture. The MIA results (Figures 1, 3) show subtle differences, making strong conclusions about relative forgetting quality difficult for methods near the 0.5 AUROC baseline.
2. **Focus on Node Classification:** Primarily evaluates unlearning for node classification. Performance trade-offs might differ for other tasks like link prediction or graph classification.
3. **Implementation vs. Algorithm:** The scalability critique focuses heavily on current *implementations* lacking batching. While reflecting the practical state, it doesn't definitively rule out that some *algorithms* could be made scalable with more engineering effort. A deeper discussion of algorithmic amenability to batching would be valuable.
4. **Novelty:** As a benchmarking paper, its novelty lies in the depth and width of its framework and insights rather than proposing new methods. So while highly valuable, I would prefer to have covered all the setups in the same framework for it to be complete.

**Questions:**

1. I would definitely like to see Cognac [1] being evaluated to retain my rating. From what I understand, it can easily be adapted to work and evaluated in this setting, even if they don't evaluate it in this setting in their paper. It might not be the state-of-the-art in this setting, but it is SOTA in a harder setting, so it still needs to be evaluated for completeness.
2. The MIA results in Figure 1 show PROJECTOR and GRAPHERASER performing poorly. Given that these methods are designed with stronger (sometimes exact) unlearning notions in mind, this seems counterintuitive. Do you have hypotheses as to why they failed the MIA test used? Does it relate to their specific architectures or the assumptions underlying the MIA?
3. You highlight the lack of batching support as a major scalability bottleneck. Based on the algorithmic principles of methods like MEGU, GIF, or GNNDelete, what are the core technical challenges in developing batch-compatible versions? Are certain paradigms inherently harder to batch than others, or is it just a matter of implementation?
4. Table 10 provides a useful summary. For MEGU, efficiency is marked 'X' due to being slower than GOLD on large graphs. However, it performs well on utility/forgetting. Could there be scenarios (frequent small deletions on medium graphs) where MEGU's overhead is acceptable given its strong utility? The guidance seems slightly harsh as it's binary (retrain vs. use X).
5. How sensitive are the efficiency results (Table 7) to implementation details? Could optimized implementations significantly change the ranking or make more methods competitive with retraining?
6. Sidenote: I'm also not sure if the paper violates margin constraints. It seems awfully dense and cluttered.

---

*[1] Kolipaka, Varshita, Akshit, Sinha, Debangan, Mishra, Sumit, Kumar, Arvindh, Arun, Shashwat, Goel, Ponnurangam, Kumaraguru. "A Cognac Shot To Forget Bad Memories: Corrective Unlearning for Graph Neural Networks." Proceedings of the 42nd International Conference on Machine Learning (ICML).*

---

> ### Author Response · Authors · 2025-11-23
> **Response to Reviewer MdZs - Part 1**
>
> We thank the reviewer for the constructive suggestions on our work. Please find below the details of how each concern has been addressed in the **revised manuscript**. All changes made in our revision are highlighted in **blue** font.
>
> ----
>
> **Q1.  I would definitely like to see Cognac \[1\] being evaluated to retain my rating. From what I understand, it can easily be adapted to work and evaluated in this setting, even if they don't evaluate it in this setting in their paper. It might not be the state-of-the-art in this setting, but it is SOTA in a harder setting, so it still needs to be evaluated for completeness.**
>
> We have now included Cognac in our benchmarking (**Tables 1, 3, 4, 5, 7, 8, 9, 11, 16, 17, 18, 24**). Key insights regarding Cognac are as follows:
>
> * **Accuracy:** Cognac is competitive in accuracy and achieves the best accuracy in 2 out of 6 datasets (Roman-E, ogbn-arxiv)
> * **Alignment with Gold:** It **falls significantly short in alignment with the GOLD model**, with poor numbers in fidelity and L2-distance in logit space. The high logit distance indicates that its predictions follow a different distribution than what retrain-from-scratch would produce.
> * **Forgetting:** While Cognac is better than Projector and GNNDelete, it ranks lower than all other methods in Inversion attack, which indicates some     leakage of topological information.
> * **Time:** Except on Roman-Empire, Cognac is **2-3× slower than retraining** (Gold standard). This overhead arises from COGNAC's alternating dual-component optimization: (1) contrastive unlearning over affected neighborhoods, and (2) decoupled ascent-descent using separate optimizers for deletion and retention sets. This design effectively doubles backpropagation cost per iteration, as opposing gradient directions (ascent vs. descent) cannot be fused into a single backward pass.
> * **Memory:** Although it consumes more memory than Gold, it scales on Reddit since batching is amenable and implemented in Cognac.
>
> **Q2. The MIA results in Figure 1 show PROJECTOR and GRAPHERASER performing poorly. Given that these methods are designed with stronger (sometimes exact) unlearning notions in mind, this seems counterintuitive. Do you have hypotheses as to why they failed the MIA test used? Does it relate to their specific architectures or the assumptions underlying the MIA?**
>
>
> * PROJECTOR removes the influence of deleted nodes by orthogonally projecting the GNN weights into a subspace that is orthogonal to the directions spanned by the features of the deleted nodes. While this achieves exact forgetting for its linear GNN, it also creates a **deterministic** and **structured** geometric signature in the weight space that potentially leads to **negative information leakage**. Specifically, an MIA adversary does not need to recover the deleted data; it only needs to train classifiers to detect when weights have been projected away from certain directions. While our MIA attack work with only the logits of the unlearned model, the logits are a function of the projected weights and becomes statistically detectable in our experiments.
>
> * GraphEraser provides exact unlearning guarantees only under a restrictive assumption: the Gold model (i.e., the retrain-from-scratch baseline) must itself be trained on the same sharded graph decomposition. For clarity, we restate the explanation already given in Sec. 3 of the paper:
>  > “Moreover, the guarantees of exactness provided by these methods hold only under the restrictive assumption that a separate GNN is trained per shard. If instead a single GNN is trained on the full graph, these guarantees no longer apply.”
>
> This requirement is necessary only when the full graph is too large to train a single GNN, which does not hold for any of our benchmark datasets. Consequently, when GraphEraser is evaluated in a realistic setting where Gold is trained on the full graph, the sharding-based exactness guarantees do not hold.

---

> > ### Author Response · Authors · 2025-11-23
> > **Response to Reviewer MdZs - Part 2**
> >
> > **Q3.  You highlight the lack of batching support as a major scalability bottleneck. Based on the algorithmic principles of methods like MEGU, GIF, or GNNDelete, what are the core technical challenges in developing batch-compatible versions? Are certain paradigms inherently harder to batch than others, or is it just a matter of implementation?**
> >
> > The challenges of developing batch-compatible versions varies dramatically by paradigm.
> >
> > First, **full-graph–dependent methods such as MEGU** rely on operator gates whose updates require visibility of the entire graph structure at once; these operators do not decompose over mini-batches, making batching inherently difficult without altering the algorithm itself. Second, **second-order methods such as GIF and IDEA**, which depend on Hessian–Vector Products (HVPs) computed over the full training objective, are also challenging to batch because HVPs couple gradients across distant nodes. When computed on subgraphs, these approximations drift significantly, making batching non-trivial and requiring algorithmic approximations rather than implementation tweaks. Finally, **trajectory-based methods such as GNNDelete**, which reconstruct or adjust training gradients influenced by the deleted nodes, face consistency issues under minibatch training: deletion-conditioned gradients derived from full-batch updates do not align with minibatch trajectories. For this paradigm, batching would again require redesigned objectives rather than improved code.
> >
> > Sharding-based methods are fully amenable to batching, but their inefficiency stems from the sharding process itself. Across these paradigms, the underlying challenge is the same: **their mathematical formulations assume full-graph or full-batch visibility**, and breaking that assumption compromises correctness. Consequently, the relative ranking between retraining and unlearning is unlikely to change without **algorithm-level innovations**, not just optimized implementations.
> >
> > **Q4.  Table 10 provides a useful summary. For MEGU, efficiency is marked 'X' due to being slower than GOLD on large graphs. However, it performs well on utility/forgetting. Could there be scenarios (frequent small deletions on medium graphs) where MEGU's overhead is acceptable given its strong utility? The guidance seems slightly harsh as it's binary (retrain vs. use X).**
> >
> > We agree that the earlier binary marking may have overstated the conclusion. While MEGU is indeed slower than Gold on large graphs, its utility and forgetting performance are consistently strong, and there _are_ realistic scenarios, such as frequent small deletions on medium-scale graphs, where the overhead may be acceptable.
> >
> > Although we attempted to convey this nuance in the accompanying text (reproduced below) of our submitted version, the table itself did not clearly reflect it. To address this, we now introduce a **new symbol “$\circ$” to denote partial satisfaction**, marking MEGU as neither fully failing nor fully meeting the efficiency criterion.
> >
> > > MEGU (on small graphs), GIF and IDEA are the only viable contenders. Among them, MEGU emerges as the most reliable choice on small datasets, consistently delivering strong results across datasets and architectures. For large-scale graphs, practitioners should continue to rely on retraining until batching-aware unlearning implementations are available and validated.
> >
> > **Q5.  How sensitive are the efficiency results (Table 7) to implementation details? Could optimized implementations significantly change the ranking or make more methods competitive with retraining?**
> >
> > Implementation-level optimizations can certainly reduce absolute running times. Techniques such as INT8 quantization, CUDA kernel fusion, or improved memory layouts can benefit _all_ methods and thus may not substantially alter their relative ordering.
> >
> > Shifting the efficiency ranking meaningfully would require **algorithmic** changes rather than low-level optimizations. This is because the dominant bottlenecks arise from the _design_ of the unlearning procedures rather than inefficient coding. For example, sharding-based approaches are expected to remain slow due to the inherent preprocessing overhead of partitioning the graph. GIF and IDEA would need fundamentally more efficient ways of computing or approximating Hessian–vector products to become competitive. MEGU would need to relax or approximate its requirement of full-graph availability within its operator gate before minibatching becomes viable. These obstacles stem from method-specific computational structures, and overcoming them requires algorithmic innovation rather than routine engineering improvements.
> >
> > **Q6.  Sidenote: I'm also not sure if the paper violates margin constraints. It seems awfully dense and cluttered.**
> >
> > Thank you for bringing this to our attention. In revising the paper, we have worked towards improved spacing, and reduced unnecessary compactness.

---

> > > ### Author Response · Authors · 2025-11-23
> > > **Response to Reviewer MdZs - Part 3**
> > >
> > > **W1.  **Limited Scope of "Forgetting":**  Relies solely on one type of MIA based on posterior shifts. Forgetting is a complex concept, and exploring other attack types (ex., attribute inference, link re-inference) or information-theoretic measures could provide a more complete picture. The MIA results (Figures 1, 3) show subtle differences, making strong conclusions about relative forgetting quality difficult for methods near the 0.5 AUROC baseline.**
> > >
> > > Thank you for the suggestion. We have now significantly expanded the scope of forgetting experiment.
> > >
> > > * **Additional discussions:** We have incorporated forgetting as an explicit requirement in our objective formulation in **Sec 2**. In addition, we have significantly expanded the discussion in both the importance of evaluating forgetting (Sec 2) and the metrics to evaluate them (Sec 4). The exact content is reproduced verbatim below for easy reference.
> > >
> > > * ### From Sec 2:
> > > > **Forgetting:** Two models may exhibit high similarity and yet differ significantly in the amount of information they retain about removed nodes. Hence, forgetting must be assessed separately from utility. Just like model similarity, there is no single all-encompassing  measure to quantify the extent of forgetting. We use the three-pronged strategy to evaluate forgetting: (i)  membership information of removed nodes in the original training set,  (ii) structural leakage caused by message passing, and (iii) confidence patterns correlated with the removed training signals. The technical details of how these aspects are assessed are discussed in Section 4.
> > >
> > > * ### From Sec 4:
> > >
> > >  >**Forgetting**
> > >  To quantify forgetting, we adopt three complementary families of attacks, each probing a different form of residual influence from the removed set.
> > >  > -   **Membership Inference Attacks (MIA).**  These evaluate whether an adversary can determine if a node belonged to the original training set.  MIAs test a basic privacy requirement, but capture only one dimension of forgetting and cannot detect structural or label-level leakage. For more details on the mathematical formulation, please refer to App. A.2.
> > >  > -   **Unlearning Inversion Attacks (UIA) [Zhang et al., 2026].**  Inversion-based attacks attempt to reconstruct _deleted edges_ from black-box access to $\widetilde{\Theta}$.  These attacks detect structural leakage caused by message passing and confidence perturbations, which MIAs do not capture.
> > >  > -   **Noisy-Labeler Attacks [Sui et al., 2025].**  We additionally evaluate whether $\widetilde{\Theta}$ acts as a “noisy labeler,” assigning removed nodes high-confidence predictions that reflect their original class memberships.  This probes _label leakage_, complementing MIAs (membership leakage) and structural leakage (UIA).
> > >
> > > * **New attacks:** Our original submission already included membership-inference attacks to assess privacy preservation (Sec 5.3). In the revised version, we have further strengthened the evaluation by incorporating two additional privacy-oriented threat models (**Section 5.3**):
> > > * **unlearning inversion attacks** \[1\] and
> > > * **noisy-labeler attacks** \[2\].
> > >
> > > Together with MIA, these attacks probe different facets of residual information, ranging from prediction-level leakage (MIA) to reconstruction of hidden representations (inversion) and label-dependent vulnerability (noisy-labeler attacks). The new results and key insights are summarized below.
> > > * MEGU, GIF, and IDEA consistently achieve AUROC ≈ 0.50 across all attacks and datasets, demonstrating robust and stable forgetting.
> > > * GNNDelete performs poorly on all three attacks, with AUROC often >0.80, indicating strong membership leakage and confirming that it is not safe for unlearning.
> > > * Projector shows clear weaknesses in MIA and inversion attacks but performs reasonably under noisy-labeler attacks. This inconsistency highlights its susceptibility to geometry-based attacks, as discussed.
> > > * GraphEraser is weak under MIA (detectable leakage), but performs well under inversion and noisy-labeler attacks. This inconsistency aligns with our earlier observation that its theoretical exactness breaks when gold is not sharded.

---

> > > > ### Author Response · Authors · 2025-11-23
> > > > **Response to MdZs- Part 4**
> > > >
> > > > ### Inversion Attack AUROC
> > > > (Closer to 0.5 is better.)
> > > > | **Method**       | **Cora**       | **Citeseer**   | **Photo**      | **Amazon-R.**  | **Roman-E.**   | **OGBN-Arxiv** |
> > > > |------------------|----------------|----------------|----------------|----------------|----------------|----------------|
> > > > | MEGU             | 0.53 ± 0.01    | 0.54 ± 0.00    | 0.53 ± 0.01    | 0.51 ± 0.00    | 0.51 ± 0.00    | 0.50 ± 0.00    |
> > > > | GIF              | 0.53 ± 0.02    | 0.54 ± 0.01    | 0.53 ± 0.01    | 0.51 ± 0.00    | 0.51 ± 0.00    | 0.50 ± 0.00    |
> > > > | IDEA             | 0.52 ± 0.01    | 0.54 ± 0.01    | 0.53 ± 0.01    | 0.51 ± 0.00    | 0.51 ± 0.00    | 0.50 ± 0.00    |
> > > > | GNNDelete        | 0.78 ± 0.05    | 0.90 ± 0.01    | 0.89 ± 0.12    | 0.95 ± 0.01    | 0.80 ± 0.01    | OOM            |
> > > > | Projector        | 0.60 ± 0.02    | 0.57 ± 0.00    | 0.53 ± 0.01    | 0.60 ± 0.01    | 0.52 ± 0.01    | 0.56 ± 0.00    |
> > > > | GraphEraser      | 0.54 ± 0.01    | 0.54 ± 0.01    | 0.53 ± 0.01    | 0.51 ± 0.00    | 0.51 ± 0.01    | 0.51 ± 0.00    |
> > > > | GUIDE            | 0.53 ± 0.00    | 0.54 ± 0.01    | 0.53 ± 0.01    | 0.51 ± 0.00    | 0.51 ± 0.00    | OOM            |
> > > > Cognac | $0.54 \pm 0.00$ | $0.58 \pm 0.01$ | $0.54 \pm 0.01$ | $0.51 \pm 0.00$ | $0.52 \pm 0.00$ | $0.50 \pm 0.00$
> > > > ETR| $0.54 \pm 0.00$| $0.53 \pm 0.01$ | $0.53 \pm 0.01$ | $0.51 \pm 0.00$ | $0.51 \pm 0.00$ | $0.50 \pm 0.00$
> > > >
> > > > ### Noisy-Labeler Attack AUROC
> > > > (Closer to 0.50 is better.)
> > > >
> > > > | **Method**       | **Cora**       | **Citeseer**   | **Photo**      | **Amazon-R.**  | **Roman-E.**   | **OGBN-Arxiv** |
> > > > |------------------|----------------|----------------|----------------|----------------|----------------|----------------|
> > > > | MEGU             | 0.50 ± 0.01    | 0.51 ± 0.02    | 0.50 ± 0.01    | 0.50 ± 0.00    | 0.50 ± 0.01    | 0.50 ± 0.00    |
> > > > | GIF              | 0.50 ± 0.01    | 0.51 ± 0.02    | 0.50 ± 0.01    | 0.51 ± 0.01    | 0.50 ± 0.01    | 0.50 ± 0.00    |
> > > > | IDEA             | 0.50 ± 0.01    | 0.50 ± 0.02    | 0.50 ± 0.01    | 0.50 ± 0.01    | 0.50 ± 0.01    | 0.50 ± 0.00    |
> > > > | GNNDelete        | 0.58 ± 0.03    | 0.59 ± 0.02    | 0.61 ± 0.06    | 0.81 ± 0.02    | 0.52 ± 0.02    | OOM            |
> > > > | Projector        | 0.50 ± 0.03    | 0.51 ± 0.04    | 0.50 ± 0.01    | 0.50 ± 0.00    | 0.50 ± 0.01    | 0.50 ± 0.00    |
> > > > | GraphEraser      | 0.49 ± 0.01    | 0.52 ± 0.01    | 0.50 ± 0.01    | 0.50 ± 0.00    | 0.50 ± 0.01    | 0.50 ± 0.00    |
> > > > | GUIDE            | 0.50 ± 0.01    | 0.51 ± 0.02    | 0.50 ± 0.01    | 0.51 ± 0.00    | 0.50 ± 0.01    | OOM            |
> > > > Cognac       | $0.51 \pm 0.00$ | $0.50 \pm 0.01$ | $0.50 \pm 0.01$ | $0.50 \pm 0.00$ | $0.50 \pm 0.01$ | $0.50 \pm 0.01$
> > > > ETR      | $0.51 \pm 0.01$ | $0.50 \pm 0.02$ | $0.50 \pm 0.01$ | $0.50 \pm 0.00$ | $0.50 \pm 0.0$ | $0.50 \pm 0.0$
> > > >
> > > > **W2.  **Focus on Node Classification:**  Primarily evaluates unlearning for node classification. Performance trade-offs might differ for other tasks like link prediction or graph classification.**
> > > >
> > > > We appreciate the reviewer’s suggestion and have now included both edge and feature unlearning in our benchmarking study (**Appendix A.3: Tables 16–18; Appendix A.4: Tables 19–21**, referred from Sec 5-first para in revised manuscript). The results are also presented below for easy acccess. These additional experiments reinforce the core trends observed earlier, with MEGU consistently providing the most robust performance across settings.
> > > >
> > > > **W3.  **Implementation vs. Algorithm:**  The scalability critique focuses heavily on current  _implementations_  lacking batching. While reflecting the practical state, it doesn't definitively rule out that some  _algorithms_  could be made scalable with more engineering effort. A deeper discussion of algorithmic amenability to batching would be valuable.**
> > > >
> > > > Implementation-level optimizations can certainly reduce absolute running times. Techniques such as INT8 quantization, CUDA kernel fusion, or improved memory layouts can benefit _all_ methods and thus may not substantially alter their relative ordering.
> > > >
> > > > Shifting the efficiency ranking meaningfully would require **algorithmic** changes rather than low-level optimizations. This is because the dominant bottlenecks arise from the _design_ of the unlearning procedures rather than inefficient coding. For example, sharding-based approaches are expected to remain slow due to the inherent preprocessing overhead of partitioning the graph. GIF and IDEA would need fundamentally more efficient ways of computing or approximating Hessian–vector products to become competitive. MEGU would need to relax or approximate its requirement of full-graph availability within its operator gate before minibatching becomes viable. These obstacles stem from method-specific computational structures, and overcoming them requires algorithmic innovation rather than routine engineering improvements.

---

> > > > > ### Author Response · Authors · 2025-11-23
> > > > > **Response to MdZs- Part 5**
> > > > >
> > > > > **W4.  **Novelty:**  As a benchmarking paper, its novelty lies in the depth and width of its framework and insights rather than proposing new methods. So while highly valuable, I would prefer to have covered all the setups in the same framework for it to be complete.**
> > > > >
> > > > > We appreciate this feedback. As discussed in our various responses, the revised version has significantly enhanced the coverage. We outline some of the key enhancements below. With these additions, we hope the reviewer would find our manuscript improved.
> > > > >
> > > > > * Inclusion of Cognac [1], ETR [2], ScaleGun [3], and CGU [4] in our benchmarks.
> > > > > * Inclusion of two new attacks: Inversion [6] and noisy labels [7]
> > > > > * Inclusion of edge and feature unlearning
> > > > > * Granular break-up of unlearning times to understand the time spent on pre-processing (which is often not reported in the literature, creating an illusion of higher efficiency than retraining, and unlearning.
> > > > >
> > > > > _\[1\] Kolipaka, Varshita, Akshit, Sinha, Debangan, Mishra, Sumit, Kumar, Arvindh, Arun, Shashwat, Goel, Ponnurangam, Kumaraguru. "A Cognac Shot To Forget Bad Memories: Corrective Unlearning for Graph Neural Networks." Proceedings of the 42nd International Conference on Machine Learning (ICML)._
> > > > >
> > > > > *[2] Zhe-Rui Yang, Jindong Han, Chang-Dong Wang, Hao Liu. “Erase then Rectify: A Training-Free Parameter Editing Approach for Cost-Effective Graph Unlearning”. AAAI 2025.*
> > > > >
> > > > > *[3] Lu Yi and Zhewei Wei. Scalable and certifiable graph unlearning: Overcoming the approximation error barrier, ICLR 2025*
> > > > >
> > > > > *[4] Eli Chien, Chao Pan, and Olgica Milenkovic. Certified graph unlearning. In NeurIPS GLFrontiers Workshop, 2022.*
> > > > >
> > > > > *[5] Jiahao Zhang, Yilong Wang, Zhiwei Zhang, Xiaorui Liu, Suhang Wang. “Unlearning Inversion Attacks for Graph Neural Networks”. WSDM 2026.*
> > > > >
> > > > > *[6] Zhihao Sui, Liang Hu, Jian Cao, Dora D. Liu, Usman Naseem, Zhongyuan Lai, Qi Zhang. “Recalling The Forgotten Class Memberships: Unlearned Models Can Be Noisy Labelers to Leak Privacy”. IJCAI, 2025.*

---

> > > > > > ### Comment · Reviewer_MdZs · 2025-11-25
> > > > > >
> > > > > > My main concern was the breadth of the experiments for a benchmarking work, and the authors have added extensive experiments during the rebuttal. This addresses most (if not all) of my concerns.
> > > > > >
> > > > > > But before I make changes to my assessment, there is a small issue in Table 1. I see MEGU and Cognac being classified as "Train-time". Having used both of the methods before, I'm fairly certain they are not train-time methods (at least by the traditional definition). Please correct me if I'm wrong or if your definition is different. It could be a genuine oversight on the authors' part, but it's quite concerning that such a fundamental detail about the methods used in a benchmarking work is wrong.

---

> > > > > > > ### Author Response · Authors · 2025-11-26
> > > > > > >
> > > > > > > We thank the reviewer for pointing out the error in Table 1. Megu and Cognac are indeed post-hoc methods, and the same correction applies to ETR. We apologize for this oversight. The mistake arose from an internal mix-up between the interpretations of *train-time unlearning* and *continual learning*: the former requires assumptions built into the original training procedure, while the latter concerns the ability to incorporate new data after unlearning. These three methods support continual training, but they do not require train-time modifications, and we mistakenly categorized them otherwise.
> > > > > > >
> > > > > > > We fully agree that such an error should not have occurred, and we appreciate the reviewer bringing it to our attention. Over the last day, we performed a thorough verification of every method to ensure consistency in Table 1. The resulting corrections and clarifications (highlighted in **brown** in the revised manuscript V2) include:
> > > > > > >
> > > > > > > * Updating Megu, Cognac, and ETR to post-hoc in the Mode column.
> > > > > > >
> > > > > > > * Clarifying that ETR uses a hybrid paradigm: it learns parameters that approximate influence gradients and then learns over these important parameter space. It therefore fits both learning and influence-function categories.
> > > > > > >
> > > > > > > * Correcting Projector from post-hoc to train-time, since it assumes customized GNN architectures to enable unlearning.
> > > > > > >
> > > > > > > * Clarifying GST: although originally designed for graph classification, its applicability to node-level continual learning is not fully clear. We conservatively list it as capable.
> > > > > > >
> > > > > > > * Explaining IDEA more precisely: its certification requires the GNN to be λ-convex. This rules out all non-linear GNNs, and even linear GNNs need not satisfy λ-convexity without architectural or loss-specific constraints. Since the method does not provide a concrete GNN–loss specification guaranteeing this property, we characterize it as not offering a general guarantee. We welcome the reviewer’s thoughts if they feel an alternative framing is more suitable.
> > > > > > >
> > > > > > > We sincerely appreciate the reviewer’s close reading and constructive engagement. We are also glad that the expanded experiments and clarifications substantially address their earlier concerns.

---

> > > > > > > > ### Comment · Reviewer_MdZs · 2025-11-26
> > > > > > > >
> > > > > > > > Given the clarifications and the breadth of relevant experiments added during the rebuttal phase, I think this is a valuable contribution to the graph unlearning domain. I'm happy to increase my score to 8.

---

> > > > > > > > > ### Author Response · Authors · 2025-11-27
> > > > > > > > >
> > > > > > > > > We are pleased to see the reviewer’s positive assessment of our work and sincerely appreciate the constructive feedback that helped strengthen the paper.

---

### Official Review · Reviewer_JbxA · 2025-10-30

**Soundness:** 2
**Presentation:** 3
**Contribution:** 2
**Rating:** 4
**Confidence:** 4

**Summary:**

The paper introduces a systematic review of recent graph unlearning techniques, and benchmarks them in line with the usual three properties used to evaluate graph unlearning: utility, forgetting, and efficiency. The authors evaluate a wide array of unlearning methods on the task of random node unlearning. They include an ablation with more targeted unlearning. In particular, they introduce new metrics to measure the utility of a model, where they also look at the fidelity, L2 distance, and parameter difference between the original and unlearnt model. Overall, I feel the work is timely, but the contributions are limited and narrow. I will discuss this further in weaknesses.

**Strengths:**

- The paper does well to create a comprehensive unlearning benchmark
- The paper is a pleasure to read, and is well-written.
- The paper goes beyond the metrics, and discusses hidden training costs (which I appreciate a lot), as well as OOM issues which most graph unlearning methods suffer from as scale increases—contrary to their claims.
- I really appreciate the strong concluding section with limitations and insights for practical deployment

**Weaknesses:**

1. Key contribution 1 (Unified benchmarking framework), is not exactly a contribution. If you look at any graph unlearning paper released in the past couple years, they all measure efficiency, utility and forgetting. This is certainly not the first comprehensive evaluation, and frankly I think is a bare minimum for any paper introducing a new method [1][2]. Therefore, I fail to see why this should be a core contribution.

2. All main experiments are performed on 1 setting: randomised node unlearning. I feel this is very limiting, and cannot in general inform broad claims about unlearning methods. Edge unlearning or feature unlearning are not even mentioned, even though most of the mentioned algorithms focus on all of them. Since the paper positions itself as a benchmark paper, I feel this is a reasonable ask. [3] is mentioned as a competing benchmark in the paper, and performs these tasks and more.

3. The reason for exclusion of [1] in the study is that "it operates on the paradigm of corrective unlearning instead of privacy preservation". I find this unsatisfactory for a number of reasons. Firstly, the experimental setting is randomised node unlearning, which by itself has nothing to do with privacy, and is used as a proxy to evaluate unlearning on a set of nodes that were asked to be deleted. From what I can see in [1], the actual procedure of unlearning is identical to other unlearning methods. Second, many of the other methods included here are included in [1] for a comparison, so why is the converse not possible?

4. Continuing from point 3, I feel [1] has a very reasonable and practical evaluation of unlearning, which is certainly better than random unlearning. Since this is a benchmark paper, would it not make sense to include various settings of attacks? A lot of unlearning papers mention that they can operate on adversarial attacks as well (see [2]), even if their main area is privacy. This is an addition to point 2. I feel the paper can benefit from including more experimental settings from both [1] and [2].

5. One point of technical novelty in the paper is the introduction of multiple levels of measuring *utility*. I agree for the need of the fidelity metric, and find that to be a nice contribution, however, I feel the other introduced metrics, the logit distance and the model-level distance, are redundant and don't offer much. I would ask the authors for a clarification on my understanding of these metrics.

    - (1) Won't a similar fidelity score imply a similar logit distance? Since the output class of the predictions is directly dependent on the logits?
    - (2) The paper says "If the parameters are close, then the intermediate activations at each GNN layer are also likely to be similar". I am not sure I can naively believe this. and would prefer some theoretical or empirical backing of this.
    - (3) I would like to ask how these metrics make a difference in the decision making guidance (since utility is mentioned as a whole there), and if they don't, what is the contribution exactly, and why are they needed?

6. I feel there should be more of a discussion on the actual "unlearning" part (forgetting). The paper discusses at length about the impact on their introduced metrics, but they are for model utility. I feel forgetting is more of an under-explored area and MIA is not the best indicator for forgetting performance. I would like the paper to have a longer discussion on forgetting. Specifically, the paper defines "The objective of unlearning is that the unlearned model Θ˜ be similar to Θ′, which is the gold standard", and this is true for both utility and forgetting metrics. However, discussion of forgetting is limited.

7. Can the authors clarify what they mean by deletion for retrain-from-scratch, specifically for RQ4? Are the nodes removed from the training set, or are they removed from the graph altogether? Both these settings can drastically change the outcome of the experiments, and warrants a careful discussion for both cases.

Overall, I feel the paper does not adequately answer any of the introduced gaps in the introduction. Evaluation ambiguity remains, Efficiency vs. retraining was the reason inexact machine unlearning methods were introduced, and I don't believe the experimental breadth is sufficient for warranting an analysis of generalisation. Thus, I am leaning towards a rejection.

[1] [Cognac](https://arxiv.org/pdf/2412.00789)

[2] [Megu](https://arxiv.org/pdf/2401.11760)

[3] [Open-GU](https://arxiv.org/pdf/2501.02728)

**Questions:**

See weaknesses.

---

> ### Author Response · Authors · 2025-11-23
> **Response to Reviewer JbxA -Part 1**
>
> We thank the reviewer for the constructive suggestions on our work. They have definitely elevated the quality our work. Please find below the changes made based on the concerns raised. All changes made in our **revised manuscript** are highlighted in **blue** font.
>
> -----
>
> **W1. Key contribution 1 (Unified benchmarking framework), is not exactly a contribution. If you look at any graph unlearning paper released in the past couple years, they all measure efficiency, utility and forgetting. This is certainly not the first comprehensive evaluation, and frankly I think is a bare minimum for any paper introducing a new method [1][2]. Therefore, I fail to see why this should be a core contribution.**
>
> We agree with the reviewer that most unlearning papers evaluate utility, efficiency, and forgetting. Our contribution differs in **how comprehensively and systematically** these dimensions are assessed, and in the fact that they are evaluated under a **single unified experimental pipeline** across all major unlearning algorithms.
>
> **Utility.**  Our evaluation goes significantly beyond the standard accuracy-based assessments used in prior work. We include:
> - Aggregate metrics (accuracy and fidelity). We note that Fidelity is an **original contribution** of this work. This measures how well the unlearned model aligns to that of retrain from scratch. Accuracy evaluates performance of unlearned model with gold labels, but not gold model (retrain).
> - **Per-instance** comparisons in logit space. To the best of our knowledge, logit space alignment with gold model has been ignored.
> - **Parameter-space** alignment with Gold.
>
> To the best of our knowledge, **no existing benchmark** evaluates unlearning utility across all these complementary perspectives. As we expand in our response to W5.1, strong performance on one utility metric does **not** imply strong performance on others, motivating the need for a richer set of diagnostics.
>
> **Efficiency.** Our efficiency analysis considers both **runtime and memory**, whereas prior work often reports only runtime. Importantly, we do not merely report aggregate numbers—we identify **root causes** of inefficiency (e.g., batching incompatibility, Hessian–Vector Product overhead, shard construction) and explain what would be required to address these issues.
> We also demonstrate that several prior papers omit preprocessing time, which can create the illusion of outperforming retraining from scratch.
>
> **Forgetting.** Our original submission included membership inference attacks.
> In the revision, we have strengthened forgetting assessment with two additional privacy-oriented evaluations:
> - **unlearning inversion attacks** [1]
> - **noisy-labeler attacks** [2]
>
> These expand the privacy stress-tests considerably beyond what is standard in the literature (**Sec 5.3**).
>
> ### **Clarification in the revised paper**
> We acknowledge that the full depth of our assessment may not have been evident from the original contribution statement.
> We have now **rewritten the contribution bullet** in the revised manuscript to clearly articulate the breadth and rigor of our unified benchmarking framework.
>
> > * **Comprehensive and diagnostic benchmarking framework.** While prior unlearning papers evaluate some subset of utility, efficiency, and forgetting, our contribution lies in conducting the f*irst holistic, multi-layered diagnostic assessment* of GNN unlearning. Concretely, we evaluate:
> > 1. *utility* across three complementary axes—aggregate metrics (accuracy, fidelity), per-instance outputs (logit-space deviations), and parameter-space proximity—revealing discrepancies missed by prior single-metric evaluations;
> >2. *forgetting quality* using a broader suite of privacy attacks, including membership inference, *unlearning inversion* [1], and *noisy-labeler* attacks [2];
> >3.  *efficiency* through both runtime and memory profiling, explicitly accounting for preprocessing overheads that prior work omits; and
> > 4. *robustness* to realistic and adversarial deletion distributions (e.g., degree-based, label-skewed), going beyond the uniform-random setups that dominate existing evaluations.
> This expanded scope enables a more reliable understanding of when unlearning works, when it fails, and why.
>
> \[1\] Jiahao Zhang, Yilong Wang, Zhiwei Zhang, Xiaorui Liu, Suhang Wang. “Unlearning Inversion Attacks for Graph Neural Networks”. WSDM 2026.
>
> \[2\] Zhihao Sui, Liang Hu, Jian Cao, Dora D. Liu, Usman Naseem, Zhongyuan Lai, Qi Zhang. “Recalling The Forgotten Class Memberships: Unlearned Models Can Be Noisy Labelers to Leak Privacy”. IJCAI, 2025.

---

> ### Author Response · Authors · 2025-11-23
> **Response to Reviewer JbxA -Part 2**
>
> **W2. All main experiments are performed on 1 setting: randomised node unlearning. I feel this is very limiting, and cannot in general inform broad claims about unlearning methods. Edge unlearning or feature unlearning are not even mentioned, even though most of the mentioned algorithms focus on all of them. Since the paper positions itself as a benchmark paper, I feel this is a reasonable ask. [3] is mentioned as a competing benchmark in the paper, and performs these tasks and more.**
>
> We appreciate the reviewer’s suggestion and have now **included both edge and feature unlearning** in our benchmarking study (**Appendix A.3: Tables 16–18; Appendix A.4: Tables 19–21**, referred from **Sec 5**-first para in revised manuscript). These additional experiments reinforce the core trends observed earlier, with MEGU consistently providing the most robust performance across settings. We also note that our original submission already evaluated node-deletion scenarios beyond uniformly random sampling. Specifically, deletions drawn from structured, non-uniform distributions such as high-degree nodes, low-degree nodes, and label-skewed subsets, as detailed in **Table 10** and **Sec 5.6** of the paper.
>
> Our initial focus on node unlearning was intentional: we **prioritized depth over breadth**. As discussed above, our study introduces several new metrics and diagnostic evaluations that substantially broaden the understanding of utility, forgetting behavior, and failure modes in node unlearning; an area that had not previously been examined at this level of granularity. These analyses were already sufficient to demonstrate that current graph unlearning methods are not yet ready for practical deployment.
>
> While prior work such as [3] includes edge and feature unlearning, we believe—respectfully—that the overall coverage of metrics, diagnostic depth, and actionable insights is more comprehensive in our benchmarking effort (as also clarified in the final paragraph of Section 1).
>
>  **W3. The reason for exclusion of [1] in the study is that "it operates on the paradigm of corrective unlearning instead of privacy preservation". I find this unsatisfactory for a number of reasons. Firstly, the experimental setting is randomised node unlearning, which by itself has nothing to do with privacy, and is used as a proxy to evaluate unlearning on a set of nodes that were asked to be deleted. From what I can see in [1], the actual procedure of unlearning is identical to other unlearning methods. Second, many of the other methods included here are included in [1] for a comparison, so why is the converse not possible?**
>
> Indeed, the methodology can be applied to any form of unlearning. However, since the objective of Cognac is corrective unlearning, we were initially concerned about whether a direct comparison would be fair. Nonetheless, we have now included Cognac in our benchmarking (**Tables 1, 3, 4, 5, 7, 8, 9, 11, 16, 17, 18, 24**). Key insights regarding Cognac are as follows:
>
> * **Accuracy:** Cognac is competitive in accuracy and achieves the best accuracy in 2 out of 6 datasets (Roman-E, ogbn-arxiv)
> * **Alignment with Gold:** It **falls significantly short in alignment with the GOLD model**, with poor numbers in fidelity and L2-distance in logit space. The high logit distance indicates that its predictions follow a different distribution than what retrain-from-scratch would produce.
> * **Forgetting:** While Cognac is better than Projector and GNNDelete, it ranks lower than all other methods in Inversion attack, which indicates some     leakage of topological information.
> * **Time:** Except on Roman-Empire, Cognac is **2-3× slower than retraining** (Gold standard). This overhead arises from COGNAC's alternating dual-component optimization: (1) contrastive unlearning over affected neighborhoods, and (2) decoupled ascent-descent using separate optimizers for deletion and retention sets. This design effectively doubles backpropagation cost per iteration, as opposing gradient directions (ascent vs. descent) cannot be fused into a single backward pass.
> * **Memory:** Although it consumes more memory than Gold, it is the only technique to scale to Reddit since batching is amenable and implemented in Cognac.

---

> > ### Author Response · Authors · 2025-11-23
> > **Response to Reviewer JbxA -Part 3**
> >
> > **W4. Continuing from point 3, I feel [1] has a very reasonable and practical evaluation of unlearning, which is certainly better than random unlearning. Since this is a benchmark paper, would it not make sense to include various settings of attacks? A lot of unlearning papers mention that they can operate on adversarial attacks as well (see [2]), even if their main area is privacy. This is an addition to point 2. I feel the paper can benefit from including more experimental settings from both [1] and [2].**
> >
> > Our original submission already included membership-inference attacks to assess privacy preservation (App A.2). In the revised version, we have further strengthened the evaluation by incorporating two additional privacy-oriented threat models (**Section 5.3**):
> > * **unlearning inversion attacks** \[1\] and
> > * **noisy-labeler attacks** \[2\].
> >
> > Together with MIA, these attacks probe different facets of residual information, ranging from prediction-level leakage (MIA) to reconstruction of topology (inversion) and label-dependent vulnerability (noisy-labeler attacks). The new results and key insights are summarized below.
> > * MEGU, GIF, and IDEA consistently achieve AUROC ≈ 0.50 across all attacks and datasets, demonstrating robust and stable forgetting.
> > * GNNDelete performs poorly on all three attacks, with AUROC often >0.80, indicating strong membership leakage and confirming that it is not safe for unlearning.
> > * Projector shows clear weaknesses in MIA and inversion attacks but performs reasonably under noisy-labeler attacks. This inconsistency highlights its susceptibility to geometry-based attacks, as discussed.
> > * GraphEraser is weak under MIA (detectable leakage), but performs well under inversion and noisy-labeler attacks. This inconsistency aligns with our earlier observation that its theoretical exactness breaks when gold is not sharded.
> >
> >
> > ### Inversion Attack AUROC
> > (Closer to 0.5 is better.)
> > | **Method**       | **Cora**       | **Citeseer**   | **Photo**      | **Amazon-R.**  | **Roman-E.**   | **OGBN-Arxiv** |
> > |------------------|----------------|----------------|----------------|----------------|----------------|----------------|
> > | MEGU             | 0.53 ± 0.01    | 0.54 ± 0.00    | 0.53 ± 0.01    | 0.51 ± 0.00    | 0.51 ± 0.00    | 0.50 ± 0.00    |
> > | GIF              | 0.53 ± 0.02    | 0.54 ± 0.01    | 0.53 ± 0.01    | 0.51 ± 0.00    | 0.51 ± 0.00    | 0.50 ± 0.00    |
> > | IDEA             | 0.52 ± 0.01    | 0.54 ± 0.01    | 0.53 ± 0.01    | 0.51 ± 0.00    | 0.51 ± 0.00    | 0.50 ± 0.00    |
> > | GNNDelete        | 0.78 ± 0.05    | 0.90 ± 0.01    | 0.89 ± 0.12    | 0.95 ± 0.01    | 0.80 ± 0.01    | OOM            |
> > | Projector        | 0.60 ± 0.02    | 0.57 ± 0.00    | 0.53 ± 0.01    | 0.60 ± 0.01    | 0.52 ± 0.01    | 0.56 ± 0.00    |
> > | GraphEraser      | 0.54 ± 0.01    | 0.54 ± 0.01    | 0.53 ± 0.01    | 0.51 ± 0.00    | 0.51 ± 0.01    | 0.51 ± 0.00    |
> > | GUIDE            | 0.53 ± 0.00    | 0.54 ± 0.01    | 0.53 ± 0.01    | 0.51 ± 0.00    | 0.51 ± 0.00    | OOM            |
> > Cognac | $0.54 \pm 0.00$ | $0.58 \pm 0.01$ | $0.54 \pm 0.01$ | $0.51 \pm 0.00$ | $0.52 \pm 0.00$ | $0.50 \pm 0.00$
> > ETR| $0.54 \pm 0.00$| $0.53 \pm 0.01$ | $0.53 \pm 0.01$ | $0.51 \pm 0.00$ | $0.51 \pm 0.00$ | $0.50 \pm 0.00$
> >
> > ### Noisy-Labeler Attack AUROC
> > (Closer to 0.50 is better.)
> >
> > | **Method**       | **Cora**       | **Citeseer**   | **Photo**      | **Amazon-R.**  | **Roman-E.**   | **OGBN-Arxiv** |
> > |------------------|----------------|----------------|----------------|----------------|----------------|----------------|
> > | MEGU             | 0.50 ± 0.01    | 0.51 ± 0.02    | 0.50 ± 0.01    | 0.50 ± 0.00    | 0.50 ± 0.01    | 0.50 ± 0.00    |
> > | GIF              | 0.50 ± 0.01    | 0.51 ± 0.02    | 0.50 ± 0.01    | 0.51 ± 0.01    | 0.50 ± 0.01    | 0.50 ± 0.00    |
> > | IDEA             | 0.50 ± 0.01    | 0.50 ± 0.02    | 0.50 ± 0.01    | 0.50 ± 0.01    | 0.50 ± 0.01    | 0.50 ± 0.00    |
> > | GNNDelete        | 0.58 ± 0.03    | 0.59 ± 0.02    | 0.61 ± 0.06    | 0.81 ± 0.02    | 0.52 ± 0.02    | OOM            |
> > | Projector  | 0.50 ± 0.03    | 0.51 ± 0.04    | 0.50 ± 0.01    | 0.50 ± 0.00    | 0.50 ± 0.01    | 0.50 ± 0.00    |
> > | GraphEraser      | 0.49 ± 0.01    | 0.52 ± 0.01    | 0.50 ± 0.01    | 0.50 ± 0.00    | 0.50 ± 0.01    | 0.50 ± 0.00    |
> > | GUIDE | 0.50 ± 0.01    | 0.51 ± 0.02    | 0.50 ± 0.01    | 0.51 ± 0.00    | 0.50 ± 0.01    | OOM            |
> > Cognac| $0.51 \pm 0.00$ | $0.50 \pm 0.01$ | $0.50 \pm 0.01$ | $0.50 \pm 0.00$ | $0.50 \pm 0.01$ | $0.50 \pm 0.01$
> > ETR      | $0.51 \pm 0.01$ | $0.50 \pm 0.02$ | $0.50 \pm 0.01$ | $0.50 \pm 0.00$ | $0.50 \pm 0.0$ | $0.50 \pm 0.0$
> >
> > \[1\] Jiahao Zhang, Yilong Wang, Zhiwei Zhang, Xiaorui Liu, Suhang Wang. “Unlearning Inversion Attacks for Graph Neural Networks”. WSDM 2026.
> >
> > \[2\] Zhihao Sui, Liang Hu, Jian Cao, Dora D. Liu, Usman Naseem, Zhongyuan Lai, Qi Zhang. “Recalling The Forgotten Class Memberships: Unlearned Models Can Be Noisy Labelers to Leak Privacy”. IJCAI, 2025.

---

> ### Author Response · Authors · 2025-11-23
> **Response to Reviewer JbxA -Part 4**
>
> **W5. One point of technical novelty in the paper is the introduction of multiple levels of measuring utility. I agree with the need for the fidelity metric and find that to be a nice contribution; however, I feel the other introduced metrics don't offer much.**
>
>  **W5.1 Won't a similar fidelity score imply a similar logit distance?**
>
> A similar fidelity score does not necessarily imply a similar logit distance. Fidelity measures agreement in predicted labels (i.e., argmax consistency), whereas logit distance measures **how close the full probability/logit distributions are to the Gold model**.
>
> Two models can produce the same predicted label for most instances (high fidelity) while their logits differ significantly (large L2 distance). This occurs because fidelity is coarse; it only checks which class has the highest logit, whereas logit distance is sensitive to the magnitude and signs of the entire logit vector.
>
> **Hypothetical example:** Consider a 3-class classification problem with Gold model logits:
>
> * Gold logits: (10, 2, 1) → predicted class = 1
>
> Now consider two unlearning methods, A and B, both of which produce the same predicted label as Gold:
>
> * Method A logits: (9.5, 2.1, 1.2)
> * Method B logits: (3.0, 2.9, 2.8)
>
> Both A and B have perfect fidelity = 1.0, but their logit distances differ dramatically. While A is very close to Gold (small L2 distance), B is far from Gold.
>
> In our experiments, **this pattern is clearly observable when comparing GIF and IDEA**. Although **both methods achieve similar fidelity scores (Table 3), IDEA exhibits a substantially larger logit distance (Table 4)**. This discrepancy persists in our edge- and feature-unlearning evaluations as well (**Tables 17–18 and Tables 20–21**). Consequently, GIF can be regarded as achieving closer distributional alignment with the Gold model.
>
>  **W5.2 The paper says "If the parameters are close, then the intermediate activations at each GNN layer are also likely to be similar". I am not sure I can naively believe this. and would prefer some theoretical or empirical backing of this.**
>
> GNN layers are Lipschitz-continuous under standard choices of aggregation and activation functions [1]. Building on this theoretical foundation, our revision now includes Lemma A.1 (reproduced verbatim below), which applies this Lipschitz continuity result to the unlearning setting.
>
> The lemma shows that if the parameters of the Gold model $,\Theta',$ and the unlearned model $,\widetilde{\Theta},$ differ by at most $\lVert \Theta' - \widetilde{\Theta} \rVert$, then the deviation in every intermediate activation vector is upper bounded by a layer-dependent Lipschitz constant times this parameter difference.
>
> In other words, while closeness in parameters does not guarantee identical activations, it does provide a theoretically grounded upper bound on how far the activations can deviate. This justifies using parameter distance as one of our diagnostic metrics.
>
> > **Lemma (Similarity of Activations Under Similar Parameters).**
> Let $f\_{\Theta}$ be a message-passing GNN with parameters $\Theta$, using standard aggregation functions (mean/sum/max) and activation functions (ReLU, sigmoid, or tanh). If $\Theta'$ denotes the parameters of the gold model and $\widetilde{\Theta}$ the parameters of an unlearned model, then the intermediate activations satisfy
> >$$
> >\big\| \mathbf{h}^{(\ell)}(\Theta') - \mathbf{h}^{(\ell)}(\widetilde{\Theta}) \big\|
> \;\le\;
> L\_{\ell}\,\|\Theta' - \widetilde{\Theta}\|,
> $$
> for some layer-dependent Lipschitz constant $L\_{\ell} > 0$. Thus, if the parameters are close, the activations at every layer are also close.
> >### Proof
> >Rauchwerger et al. (2025) show that message-passing GNN layers are **Lipschitz continuous**. Let $L\_\ell$ denote the Lipschitz constant of layer $\ell$.
> For a generic MPNN layer of the form
> $$
> \mathbf{h}^{(\ell+1)}\_v=\phi\left(
> \mathbf{W}\_\ell \cdot \mathrm{AGG}\big(\{\mathbf{h}^{(\ell)}\_u : u \in \mathcal{N}(v)\}\big)
> \right),
> $$
> the Lipschitz constant satisfies
> $$
> L\_\ell = L\_{\mathrm{AGG}} \, \|\mathbf{W}\_\ell\|_2,
> $$
> where:
> >- $L\_{\mathrm{AGG}} = 1$ for **mean** and **max** aggregation,
> >- $L\_{\mathrm{AGG}} \le d\_{\max}$ for **sum** aggregation,
> >- ReLU, $\tanh$, and sigmoid are at most **1-Lipschitz**, so they do not increase the bound.
> Because a **composition of Lipschitz functions is Lipschitz**, the entire $L$-layer GNN satisfies:
> $$
> \big\|\mathbf{h}^{(L)}(\Theta') - \mathbf{h}^{(L)}(\widetilde{\Theta})\big\|
> \;\le\;
> \left( \prod\_{\ell=0}^{L-1} L\_\ell \right)
> \|\Theta' - \widetilde{\Theta}\|.
> $$
> Thus, if the parameters of the Gold model $\Theta'$ and the unlearned model $\widetilde{\Theta}$ are close, then their intermediate activations at every layer are also guaranteed to be close, up to the (typically small) product of per-layer Lipschitz constants. Modern GNNs are shallow ($L \le 3$), so this product does not blow up in practice.

---

> > ### Author Response · Authors · 2025-11-23
> > **Response to Reviewer JbxA -Part 5**
> >
> > **W5.3 how these metrics make a difference in the decision making guidance**
> >
> > Current benchmarking efforts overwhelmingly quantify utility using only accuracy, which provides a coarse view of the unlearned model's predictive accuracy. However, the gold model is the one retrained from scratch after deleting data, which is precisely the requirement mandated in settings such as GDPR. Thus, a meaningful utility evaluation should reflect how close an unlearned model is to this gold model, not merely how accurate it is on a test split.
> >
> > This motivated us to introduce fidelity and logit-distance as two complementary and previously underexplored metrics. Our results show that these metrics can meaningfully diverge: for example, GIF and IDEA achieve similar fidelity, yet exhibit substantially different logit distances. This demonstrates that choosing an unlearning method based solely on accuracy (or even solely on fidelity) can be misleading, and that a nuanced assessment of utility is necessary.
> >
> > More broadly, how model similarity should be measured remains an open question. Different stakeholders may reasonably prioritize aggregate fidelity, stability in logits, or closeness in parameter space. Our contribution is therefore not to advocate a single “best’’ metric, but to provide a set of principled metrics grounded in different philosophies of model similarity, and to highlight how these perspectives can lead to different conclusions in practice.
> >
> > **W6. I would like the paper to have a longer discussion on forgetting. Specifically, the paper defines "The objective of unlearning is that the unlearned model $\tilde{\Theta}$ be similar to $\Theta'$, which is the gold standard", and this is true for both utility and forgetting metrics. However, discussion of forgetting is limited.**
> >
> > We agree with the reviewer. Forgetting is a necessary condition as also highlighted in our decision-making guidance (Table 10). To address this, we have incorporated forgetting as an explicit requirement in our objective formulation in **Sec 2**. In addition, we have significantly expanded the discussion in both the importance of evaluating forgetting (Sec 2) and the metrics to evaluate them (Sec 4). The exact content is reproduced verbatim below for easy reference. As already noted in our response to W4, we have also added two more attacks to evaluate forgetting/privacy related performance of unlearning algorithms.
> >
> > ### From Sec 2
> > > **Forgetting:** Two models may exhibit high similarity and yet differ significantly in the amount of information they retain about removed nodes. Hence, forgetting must be assessed separately from utility. Just like model similarity, there is no single all-encompassing  measure to quantify the extent of forgetting. We use the three-pronged strategy to evaluate forgetting: (i)  membership information of removed nodes in the original training set,  (ii) structural leakage caused by message passing, and (iii) confidence patterns correlated with the removed training signals. The technical details of how these aspects are assessed are discussed in Section 4.
> >
> > ### From Sec 4:
> >
> > > **Forgetting**
> > To quantify forgetting, we adopt three complementary families of attacks, each probing a different form of residual influence from the removed set.
> > >-   **Membership Inference Attacks (MIA).** These evaluate whether an adversary can determine if a node belonged to the original training set. MIAs test a basic privacy requirement, but capture only one dimension of forgetting and cannot detect structural or label-level leakage.  For more details on the mathematical formulation, please refer to App. A.2.
> > >-   **Unlearning Inversion Attacks (UIA) [Zhang et al., 2026].** Inversion-based attacks attempt to reconstruct _deleted edges_ from black-box access to $\widetilde{\Theta}$. These attacks detect structural leakage caused by message passing and confidence perturbations, which MIAs do not capture.
> > >-   **Noisy-Labeler Attacks [Sui et al., 2025].**  We additionally evaluate whether $\widetilde{\Theta}$ acts as a “noisy labeler,” assigning removed nodes high-confidence predictions that reflect their original class memberships. This probes _label leakage_, complementing MIAs (membership leakage) and structural leakage (UIA).
> >
> > The results of these attacks are discussed in our response to W4 above.
> >
> >  **W7. Can the authors clarify what they mean by deletion for retrain-from-scratch, specifically for RQ4? Are the nodes removed from the training set, or are they removed from the graph altogether? Both these settings can drastically change the outcome of the experiments, and warrants a careful discussion for both cases.**
> >
> > The nodes are only removed from the training set. We have clarified this now in Sec 5.5.

---

> > > ### Comment · Reviewer_JbxA · 2025-11-25
> > >
> > > I appreciate the very detailed rebuttal by the authors. The responses are of high quality, and cover almost all the initial weaknesses identified in the initial draft. I am impressed by the effort put into the rebuttal. The current version of the paper seems much more comprehensive and complete as evaluation of graph unlearning. Considering the major improvements, I am happy to bump my score to 8.
> > >
> > > My final suggestion to the authors would be to reorganise findings in the updated version and bring forward some material from the appendix (eg: edge unlearning), wherever it makes sense.

---

> > > > ### Author Response · Authors · 2025-11-25
> > > > **Thank you to Reviewer JbxA for the positive feedback**
> > > >
> > > > We appreciate the positive feedback on our revised manuscript and are grateful for the increase in the rating of our work. We have noted the suggestion to move edge unlearning into the main section. One option is to broaden the scope of RQ5, where we currently study node unlearning under non-random deletion patterns. This section can be expanded to include both edge and feature unlearning, with part of the edge unlearning results presented in the main text and the remaining material placed in the appendix.
> > > >
> > > > If this direction is acceptable, we propose to make these structural changes after receiving feedback from the remaining reviewers. They may offer similar suggestions and updating the organization now would require us to revise multiple section and table references across our other rebuttal responses.
> > > >
> > > > regards,
> > > >
> > > > Authors

---

### Official Review · Reviewer_hgen · 2025-10-31

**Soundness:** 3
**Presentation:** 3
**Contribution:** 3
**Rating:** 6
**Confidence:** 3

**Summary:**

This paper presents a systematic benchmark for Graph Neural Network (GNN) unlearning methods, evaluating existing approaches across three core desiderata: efficiency relative to retraining from scratch, utility preservation across parameter/logit/performance spaces, and forgetting quality via membership inference attacks. Through extensive experiments on seven datasets (Cora, Citeseer, Photo, Amazon-Ratings, Roman-Empire, OGBN-Arxiv, and Reddit), the authors evaluate eight unlearning algorithms and find that most techniques fail to scale to large graphs due to memory constraints and lack batching-aware implementations. The benchmark reveals that while methods like MEGU achieve strong utility preservation, they often fail the efficiency criterion by being slower than retraining, and that certified unlearning approaches are restricted to impractical linear GNN architectures. The paper concludes that retraining from scratch remains the most viable option for large-scale graphs, calling for fundamental redesigns of unlearning algorithms to support batching.

**Strengths:**

1. **Comprehensive multi-level evaluation framework**. The paper introduces a rigorous assessment methodology that evaluates unlearning quality at four hierarchical levels (parameter space via L2 distance, logit space via distributional divergence, per-instance fidelity, and aggregate accuracy metrics), which is substantially more thorough than prior work that relied solely on aggregate performance metrics. This multi-level analysis successfully reveals subtle discrepancies, such as IDEA achieving reasonable fidelity (0.92 on Cora in Table 4) while exhibiting large logit divergence (18.03 on Cora in Table 5), demonstrating that aggregate metrics alone are insufficient for assessing unlearning quality.

2. **Critical practical insights with actionable guidance**. The paper goes beyond empirical results to provide concrete decision-making guidelines (Table 10) by systematically comparing all methods against three essential pillars and revealing critical overlooked issues. Specifically, the work exposes widespread runtime under-reporting in existing literature (Section 5.3 discusses how MEGU excludes adjacency matrix conversion, GIF/IDEA omit neighborhood identification, and GNNDELETE ignores edge mask preprocessing), and demonstrates that most methods fail on large-scale graphs where unlearning is most needed (all methods OOM on Reddit except GOLD as shown in Tables 7-8), highlighting a fundamental gap between theoretical unlearning research and practical deployment requirements.

3. **Insightful analysis of theoretical versus practical guarantees**. The paper provides valuable critical examination of certified unlearning methods (Section 3, lines 160-171), clearly articulating why approaches like PROJECTOR, CGU, and SCALEGUN have limited practical applicability despite theoretical exactness guarantees. Specifically, it explains that these methods are "restricted to a very narrow setting" requiring custom linear GNNs, binary classification for CGU/SCALEGUN, and assumptions of tractable low-rank Hessians that "break the efficient update scheme" for non-linear architectures.

**Weaknesses:**

1. **Limited forgetting evaluation and missing adversarial analysis**. The forgetting assessment relies solely on a simple membership inference attack measuring L2 distance between pre- and post-unlearning posteriors (Appendix A.2), which may not detect sophisticated privacy leaks

2. **Incomplete coverage of recent methods and missing evaluation against state-of-the-art attacks**. While the paper claims to provide "the first comprehensive evaluation of GNN unlearning" (page 2, line 67), it omits several recent and promising approaches that address the scalability and efficiency concerns raised in the paper. Notably, Zhang's "Graph unlearning with efficient partial retraining" (WWW 2024) [1] and Yang et al.'s "Erase then Rectify: A Training-Free Parameter Editing Approach" (AAAI 2025) [2] may also require evaluation in this benchmark. Furthermore, the forgetting evaluation in Section 5.4 and Appendix A.2 uses only basic membership inference attacks, ignoring recent attack methods such as the unlearning inversion attacks by Zhang et al. [3] and the noisy labeler attacks by Sui et al. [4] that specifically target unlearned models and can leak class membership information. Without evaluating against these recent baselines and attacks, the benchmark's claim to provide comprehensive guidance (as stated in the contributions on page 2, lines 72-78) is weakened, particularly since these newer methods may offer better efficiency-utility tradeoffs and the newer attacks may reveal vulnerabilities missed by the simple L2-based MIA employed in the current evaluation.


### References
[1] Jiahao Zhang. “Graph unlearning with efficient partial retraining”. WWW 2024.

[2] Zhe-Rui Yang, Jindong Han, Chang-Dong Wang, Hao Liu. “Erase then Rectify: A Training-Free Parameter Editing Approach for Cost-Effective Graph Unlearning”. AAAI 2025.

[3] Jiahao Zhang, Yilong Wang, Zhiwei Zhang, Xiaorui Liu, Suhang Wang. “Unlearning Inversion Attacks for Graph Neural Networks”. WSDM 2026.

[4] Zhihao Sui, Liang Hu, Jian Cao, Dora D. Liu, Usman Naseem, Zhongyuan Lai, Qi Zhang. “Recalling The Forgotten Class Memberships: Unlearned Models Can Be Noisy Labelers to Leak Privacy”. IJCAI, 2025.

**Questions:**

1. Could the authors justify why the current evaluation is sufficient to assess GDPR compliance?

2. How would the conclusions in Table 10 and Section 6 change if these methods failed under stronger attack scenarios?

---

> ### Author Response · Authors · 2025-11-23
> **Response to Reviewer hgen - Part 1**
>
> We thank the reviewer for the constructive feedback on our work. Please find below details of how they have been incorporated in our **revised manuscript**. All changes made in our revision are highlighted in **blue** font.
>
> ---
>
> **W1.The forgetting assessment relies solely on a simple membership inference attack measuring L2 distance between pre- and post-unlearning posteriors (Appendix A.2), which may not detect sophisticated privacy leaks.**
>
> In the revised version, we have further strengthened the evaluation by incorporating two additional privacy-oriented threat models (**Section 5.3**):
> * **unlearning inversion attacks** \[1\] and
> * **noisy-labeler attacks** \[2\].
>
> Together with MIA, these attacks probe different facets of residual information, ranging from prediction-level leakage (MIA) to reconstruction of topology (inversion) and label-dependent vulnerability (noisy-labeler attacks). The new results and key insights are summarized below.
> * MEGU, GIF, and IDEA consistently achieve AUROC ≈ 0.50 across all attacks and datasets, demonstrating robust and stable forgetting.
> * GNNDelete performs poorly on all three attacks, with AUROC often >0.80, indicating strong membership leakage and confirming that it is not safe for unlearning.
> * Projector shows clear weaknesses in MIA and inversion attacks but performs reasonably under noisy-labeler attacks. This inconsistency highlights its susceptibility to geometry-based attacks, as discussed.
> * GraphEraser is weak under MIA (detectable leakage), but performs well under inversion and noisy-labeler attacks. This inconsistency aligns with our earlier observation that its theoretical exactness breaks when gold is not sharded.
>
>
> ### Inversion Attack AUROC
> (Closer to 0.5 is better.)
> | **Method**       | **Cora**       | **Citeseer**   | **Photo**      | **Amazon-R.**  | **Roman-E.**   | **OGBN-Arxiv** |
> |------------------|----------------|----------------|----------------|----------------|----------------|----------------|
> | MEGU             | 0.53 ± 0.01    | 0.54 ± 0.00    | 0.53 ± 0.01    | 0.51 ± 0.00    | 0.51 ± 0.00    | 0.50 ± 0.00    |
> | GIF              | 0.53 ± 0.02    | 0.54 ± 0.01    | 0.53 ± 0.01    | 0.51 ± 0.00    | 0.51 ± 0.00    | 0.50 ± 0.00    |
> | IDEA             | 0.52 ± 0.01    | 0.54 ± 0.01    | 0.53 ± 0.01    | 0.51 ± 0.00    | 0.51 ± 0.00    | 0.50 ± 0.00    |
> | GNNDelete        | 0.78 ± 0.05    | 0.90 ± 0.01    | 0.89 ± 0.12    | 0.95 ± 0.01    | 0.80 ± 0.01    | OOM            |
> | Projector        | 0.60 ± 0.02    | 0.57 ± 0.00    | 0.53 ± 0.01    | 0.60 ± 0.01    | 0.52 ± 0.01    | 0.56 ± 0.00    |
> | GraphEraser      | 0.54 ± 0.01    | 0.54 ± 0.01    | 0.53 ± 0.01    | 0.51 ± 0.00    | 0.51 ± 0.01    | 0.51 ± 0.00    |
> | GUIDE            | 0.53 ± 0.00    | 0.54 ± 0.01    | 0.53 ± 0.01    | 0.51 ± 0.00    | 0.51 ± 0.00    | OOM            |
> Cognac | $0.54 \pm 0.00$ | $0.58 \pm 0.01$ | $0.54 \pm 0.01$ | $0.51 \pm 0.00$ | $0.52 \pm 0.00$ | $0.50 \pm 0.00$
> ETR| $0.54 \pm 0.00$| $0.53 \pm 0.01$ | $0.53 \pm 0.01$ | $0.51 \pm 0.00$ | $0.51 \pm 0.00$ | $0.50 \pm 0.00$
>
> ### Noisy-Labeler Attack AUROC
> (Closer to 0.50 is better.)
>
> | **Method**       | **Cora**       | **Citeseer**   | **Photo**      | **Amazon-R.**  | **Roman-E.**   | **OGBN-Arxiv** |
> |------------------|----------------|----------------|----------------|----------------|----------------|----------------|
> | MEGU             | 0.50 ± 0.01    | 0.51 ± 0.02    | 0.50 ± 0.01    | 0.50 ± 0.00    | 0.50 ± 0.01    | 0.50 ± 0.00    |
> | GIF              | 0.50 ± 0.01    | 0.51 ± 0.02    | 0.50 ± 0.01    | 0.51 ± 0.01    | 0.50 ± 0.01    | 0.50 ± 0.00    |
> | IDEA             | 0.50 ± 0.01    | 0.50 ± 0.02    | 0.50 ± 0.01    | 0.50 ± 0.01    | 0.50 ± 0.01    | 0.50 ± 0.00    |
> | GNNDelete        | 0.58 ± 0.03    | 0.59 ± 0.02    | 0.61 ± 0.06    | 0.81 ± 0.02    | 0.52 ± 0.02    | OOM            |
> | Projector        | 0.50 ± 0.03    | 0.51 ± 0.04    | 0.50 ± 0.01    | 0.50 ± 0.00    | 0.50 ± 0.01    | 0.50 ± 0.00    |
> | GraphEraser      | 0.49 ± 0.01    | 0.52 ± 0.01    | 0.50 ± 0.01    | 0.50 ± 0.00    | 0.50 ± 0.01    | 0.50 ± 0.00    |
> | GUIDE            | 0.50 ± 0.01    | 0.51 ± 0.02    | 0.50 ± 0.01    | 0.51 ± 0.00    | 0.50 ± 0.01    | OOM            |
> Cognac       | $0.51 \pm 0.00$ | $0.50 \pm 0.01$ | $0.50 \pm 0.01$ | $0.50 \pm 0.00$ | $0.50 \pm 0.01$ | $0.50 \pm 0.01$
> ETR      | $0.51 \pm 0.01$ | $0.50 \pm 0.02$ | $0.50 \pm 0.01$ | $0.50 \pm 0.00$ | $0.50 \pm 0.0$ | $0.50 \pm 0.0$
>
> \[1\] Jiahao Zhang, Yilong Wang, Zhiwei Zhang, Xiaorui Liu, Suhang Wang. “Unlearning Inversion Attacks for Graph Neural Networks”. WSDM 2026.
>
> \[2\] Zhihao Sui, Liang Hu, Jian Cao, Dora D. Liu, Usman Naseem, Zhongyuan Lai, Qi Zhang. “Recalling The Forgotten Class Memberships: Unlearned Models Can Be Noisy Labelers to Leak Privacy”. IJCAI, 2025.

---

> > ### Author Response · Authors · 2025-11-23
> > **Response to Reviewer hgen - Part 2**
> >
> > **W2. Incomplete coverage of recent methods and missing evaluation against state-of-the-art attacks. Zhang's "Graph unlearning with efficient partial retraining" (WWW 2024) \[1\] and Yang et al.'s "Erase then Rectify: A Training-Free Parameter Editing Approach" (AAAI 2025) \[2\] may also require evaluation in this benchmark. Furthermore, the forgetting evaluation in Section 5.4 and Appendix A.2 uses only basic membership inference attacks, ignoring recent attack methods such as the unlearning inversion attacks by Zhang et al. \[3\] and the noisy labeler attacks by Sui et al. \[4\]...**
> >
> >  As discussed above, we have now added the two new attack methods suggested by the reviewer.
> >
> > To improve coverage of unlearning algorithms, we have now added two state-of-the-art unlearning methods: namely **Cognac** [ICML 2025] and **ETR** [AAAI 2025] (as suggested by the reviewer). They have been evaluated across all our experiments and analysis including characterization (Table 1), accuracy (Table 3), fidelity (Table 4), logit similarity (Table 5), time consumption (Table 7), Memory footprint (Table 8), decision making (Table 11), edge unlearning (Table 16), inversion attack (Table 9), and  noisy-label attack (Table 24). Key insights are as follows:
> >
> > * **Accuracy:** While they achieve the best accuracy in some datasets, they are not consistently superior to other methods.
> > * **Alignment with Gold:** Both techniques **fall significantly short in alignment with the GOLD model**, with poor numbers in fidelity and L2-distance in logit space. The high logit distance indicates that its predictions follow a different distribution than what retrain-from-scratch would produce. When both accuracy and alignment metrics are taken into account MEGU retains its position as the most robust method.
> > * **Forgetting:** While Cognac is significantly better than Projector and GNNDelete, it ranks marginally lower than most other methods in Inversion attack, which indicates some leakage of topological information. ETR performs well and do not show any signs of information leakage.
> > * **Time:** Except on Roman-Empire, Cognac is **2-3× slower than retraining** (Gold standard). This overhead arises from COGNAC's alternating dual-component optimization: (1) contrastive unlearning over affected neighborhoods, and (2) decoupled ascent-descent using separate optimizers for deletion and retention sets. This design effectively doubles backpropagation cost per iteration, as opposing gradient directions (ascent vs. descent) cannot be fused into a single backward pass. **ETR, on the other hand, is the only method to scale on Reddit with faster unlearning time than retrain.** Hence, we now recommend ETR as the most suitable algorithm for large datasets. Nonetheless, ETR is not perfect since alignment with Gold is not ETR is not perfect since alignment with Gold is significantly inferior to MEGU.
> > * **Memory:** Both Cognac and ETR support batching and are therefore exlcusive in their ability to scale to Reddit. We, however, note that Cognac consumes more memory than Gold.
> >
> > **Q1. Could the authors justify why the current evaluation is sufficient to assess GDPR compliance?**
> >
> > GDPR’s right to erasure requires that an individual’s data be removed such that the resulting system behaves as if the data had never been collected. In principle, this implies retraining from scratch after deleting the relevant samples. Since full retraining is often computationally prohibitive, unlearning methods aim to approximate this outcome by removing the model’s dependence on the deleted data without retraining while achieving comparable predictive quality at significantly lower computational cost.
> >
> > Our benchmarking evaluates these core requirements: whether an unlearning method removes residual memory of deleted samples (privacy attacks), preserves model utility, and offers efficiency relative to retraining. Together, these dimensions assess the technical prerequisites for GDPR-aligned forgetting while also capturing the practical constraints under which unlearning would be deployed. To further strengthen the privacy evaluation, we have incorporated the additional privacy metrics suggested by the reviewer into the revised manuscript.
> >
> > **Q2. How would the conclusions in Table 10 and Section 6 change if these methods failed under stronger attack scenarios?**
> >
> > Forgetting is a necessary condition for any practical unlearning method: if a model continues to leak information about deleted data under more powerful privacy attacks, then, regardless of its accuracy or speed, it cannot be considered a viable unlearning solution.

---

### Official Review · Reviewer_hfMm · 2025-10-31

**Soundness:** 2
**Presentation:** 3
**Contribution:** 2
**Rating:** 4
**Confidence:** 4

**Summary:**

This paper introduces a systematic benchmark for graph unlearning, organized around three desiderata: efficiency, utility, and forgetting. It evaluates multiple families of unlearning methods across diverse datasets and deletion scenarios, comparing them against a retrained “gold” model. The study reports that while current techniques can sometimes approximate retraining, most are not yet practical for large graphs, and it offers actionable guidance for when to prefer unlearning vs. retraining, plus an open-source implementation.

**Strengths:**

1.	Clear, multi-level evaluation framework: The work formalizes efficiency/utility/forgetting pillars and defines metrics including fidelity and logit distance beyond accuracy, improving diagnostic resolution. This enhances experimental rigor and comparability. Forgetting measured via MIAs tailored to unlearning, aligning with privacy goals. This matters for trustworthiness.
2.	Broad empirical coverage with actionable insights: This paper applies seven datasets spanning homophily/heterophily and scale, which supports generality claims.
3.	Effective supplementary material: The appendix provides details and support the main body findings.

**Weaknesses:**

1.	Limited inclusion of certified methods in main comparisons: Certified approaches are discussed but excluded from benchmarking due to constraints (Sec. 5.1), reducing completeness of the “landscape” under the proposed metrics.
2.	Runtime accounting and preprocessing costs: Although the paper critiques under-reporting, its own measurements still aggregate components differently across methods; explicit timing breakdowns (graph construction, neighbourhood extraction, IF computations, etc.) are not standardized. Some claims (e.g., batching incompatibility specifics) are argued qualitatively without per-method ablations quantifying what batching would save or lose.
3.	Lack of theoretical evidence: The paper does not propose or establish any new theoretical results. It would be beneficial to include theoretical foundations or analyses to support the discussion of existing unlearning algorithms.
4.	Privacy metrics: It is advisable to incorporate privacy metrics into the framework, as privacy is a primary motivation for unlearning.

**Questions:**

See weaknesses.

---

> ### Author Response · Authors · 2025-11-23
> **Response to Reviewer hfMm - Part 1**
>
> We thank the reviewer for the constructive suggestions on our work. Please find below the details of how each concern has been addressed in the **revised manuscript**. All changes made in our revision are highlighted in **blue** font.
>
> ----
>
>  **W1. Limited inclusion of certified methods in main comparisons: Certified approaches are discussed but excluded from benchmarking due to constraints (Sec. 5.1), reducing completeness of the “landscape” under the proposed metrics.**
>
> We excluded certified unlearning approaches in the initial submission because **they rely on custom-designed linear GNN architectures that are incompatible with the standard GNNs** used by other unlearning methods, preventing a fair comparison. More critically, they support **only binary node classification**. For multi-class datasets, it reduces the problem to a one-vs-rest setting. Consequently, the resulting accuracy, fidelity, and logit-distance metrics are **not directly comparable** to the numbers produced by other unlearning algorithms that operate natively in the multi-class regime.
>
> Nonetheless, to fully address the reviewer’s concerns, we have now added ScaleGun [ICLR 2025] and CGU [NeurIPS GLFrontiers
> Workshop, 2022]. The corresponding results are included in **Appendix A.5**. The performance table is also presented below for easy access. In ScaleGun, on Cora, Citeseer, Photo, and OGBN-Arxiv, the unlearned accuracy remains close to the Gold model and fidelity is reasonably high. However, on heterophilous datasets Amazon-Ratings and Roman-Empire, accuracy drops substantially and fidelity collapses, indicating unstable performance.
>
> ### ScaleGun
> | **Dataset**         | **Orig. Acc** | **Gold Acc** | **Unlearned Acc** | **Fidelity** | **L2 Logit Distance** |
> |---------------------|--------------|--------------|--------------------|--------------|------------------|
> | **Cora**            | 81.7         | 82.2         | 81.5               | 92.6         | 0.51             |
> | **Citeseer**        | 75.9         | 75.48        | 74.7               | 93.8         | 0.602            |
> | **Photo**           | 50.9         | 50.7         | 50.7               | 99.0         | 0.12             |
> | **Amazon-Ratings**  | 32.2         | 31.7         | 20.6               | 5.8          | 0.03             |
> | **Roman-Empire**    | 42.7         | 42.6         | 15.3               | 5.6          | 0.54             |
> | **OGBN-Arxiv**      | 40.64        | 40.69        | 40.67              | 99.0         | 0.192            |
>
> ### CGU
>
> | Dataset        | Orig. Acc | Gold Acc | Unlearnt Acc | Fidelity | Logit Dist. |
> |----------------|-----------|----------|--------------|----------|-------------|
> | Cora           | 87.4      | 87.6     | 87.4         | 99       | 0.12        |
> | Citeseer       | 75.9      | 75.9     | 75.1         | 97.1     | 0.144       |
> | Photos         | 61.9      | 62.2     | 62.2         | 99.8     | 0.2         |
> | Amazon-ratings | 35.0      | 34.7     | 35.2         | 96.3     | 0.05        |
> | Roman-empire   | 28.4      | 28.3     | 28.4         | 97.1     | 0.06        |
> | ogbn-arxiv     | OOT       | OOT      | OOT          | OOT      | OOT         |

---

> > ### Author Response · Authors · 2025-11-23
> > **Response to Reviewer hfMm - Part 2**
> >
> > **W2. Runtime accounting and preprocessing costs: Although the paper critiques under-reporting, its own measurements still aggregate components differently across methods; explicit timing breakdowns (graph construction, neighbourhood extraction, IF computations, etc.) are not standardized. Some claims (e.g., batching incompatibility specifics) are argued qualitatively without per-method ablations quantifying what batching would save or lose.**
> >
> > We believe there is a misunderstanding between _standardizing preprocessing_ and _accounting for preprocessing_. The preprocessing steps required by different unlearning algorithms are inherently heterogeneous—e.g., shard construction in GraphEraser and Guide vs. subgraph extraction and neighborhood enumeration in MEGU/GNNDelete. Because these steps are algorithm-specific and not comparable in structure or purpose, “standardizing” pre-processing is not feasible.
> >
> > Our critique is therefore not about standardizing preprocessing pipelines, but about **under-reporting of preprocessing time** in prior works. Several papers measure only the core optimization/update steps while omitting substantial overheads such as subgraph construction, neighborhood identification, sparse-tensor conversion, and shard pruning. To address this gap, our revision now reports:
> >
> > * **Total runtime** along with the **percentage contributed by preprocessing** (Table 7),
> > * **Unlearning-only time** (Table 21), and
> > * **Preprocessing time** (Table 22).
> >
> > For convenience, we also reproduce Table 7 below. Our results show that for many techniques, preprocessing alone accounts for more than 50% of the total cost, often making them slower than retraining from scratch. This directly challenges the central premise of unlearning and highlights the need for transparent running time accounting.
> >
> > The most extreme case is seen in MEGU with an unusually high fraction of time attributed to preprocessing (≈100%). In the official implementation, the “unlearning time” includes only the loss-optimization phase, which is extremely fast because MEGU operates on cached pre-processed data and performs only a few lightweight gradient steps. In contrast, the expensive operations, such as extracting k-hop neighborhoods, constructing adjacency subgraphs, converting them to GPU-ready sparse tensors, and copying them to the GPU, are all performed prior to optimization and therefore counted as preprocessing. As a result, the preprocessing dominates the total running time.
> >
> > | Dataset            | Gold        | Megu          | Gif           | Idea          | GnnDelete     | GraphEraser        | Guide               | Projector        |
> > |-------------------|-------------|---------------|---------------|---------------|----------------|---------------------|----------------------|-------------------|
> > | Cora              | 0.65 (41%)  | 0.61 (100%)   | 0.39 (31%)    | 0.38 (45%)    | 0.60 (23%)     | 22.93 (46%)    | 77.66 (46%)      | 1.69 (0%)   |
> > | Citeseer          | 0.66 (41%)  | 0.63 (98%)    | 0.45 (18%)    | 0.48 (52%)    | 0.62 (23%)     | 23.20 (44%)     | 81.05 (45%)   | 10.57 (0%) |
> > | Photo             | 0.90 (44%)  | 0.51 (100%)   | 0.37 (78%)    | 0.40 (92%)    | 0.87 (28%)     | 30.02 (48%)     | 919.00 (73%)    | 0.47 (0%)         |
> > | Amazon-Ratings    | 1.70 (57%)  | 2.50 (100%) | 0.74 (54%)  | 0.47 (94%)    | 1.24 (35%)     | 28.70 (44%)    | 782.20 (68%)   | 0.19 (0%)         |
> > | Roman-Empire      | 1.77 (62%)  | 2.28 (100%) | 0.63 (56%)  | 0.54 (46%)    | 1.08 (32%)     | 27.13 (50%)     | 442.20 (75%)   | 0.32 (0%)         |
> > | OGBN-Arxiv        | 9.90 (49%)  | 20.49 (100%) | 4.79 (60%) | 5.31 (68%)    | OOM        | 116.00 (53%)    | OOM           | 0.30 (0%)         |
> > | Reddit            | 96.73       | OOM| OOM       | OOM | OOM|OOT      | OOM      | OOM|

---

> > > ### Author Response · Authors · 2025-11-23
> > > **Response to Reviewer hfMm - Part 3**
> > >
> > > **W3. Lack of theoretical evidence: The paper does not propose or establish any new theoretical results. It would be beneficial to include theoretical foundations or analyses to support the discussion of existing unlearning algorithms.**
> > >
> > > Our submission is to the **ICLR Datasets and Benchmarks Track**, where the primary objective is to provide a rigorous, reproducible, and diagnostic empirical evaluation, rather than to introduce new theoretical results. Accordingly, our focus is on establishing a unified and trustworthy empirical basis for comparing graph unlearning methods.
> > >
> > > That said, we carefully revisited the manuscript to identify where additional theoretical grounding would genuinely enhance clarity. In the revision, we now **include a formal connection between parameter similarity and activation similarity (Sec. 2, Lemma A.1)**. This result leverages the fact that message-passing GNN layers are Lipschitz continuous. Consequently, if the parameters of the unlearned model are close to those of the Gold (retrain-from-scratch) model, the resulting layer-wise activations are also provably close. This provides theoretical support for using parameter-space metrics as part of our diagnostic toolkit.
> > >
> > > We would also like to emphasize that our empirical findings reveal a mismatch between existing theory and what holds in practice. Methods with the strongest formal guarantees (e.g., CGU, ScaleGUN, Projector) apply only to restricted linear GNNs, and their guarantees do not transfer to expressive architectures or large-scale datasets. Empirically, these methods often underperform or fail under realistic conditions.

---

> > > > ### Author Response · Authors · 2025-11-23
> > > > **Response to Reviewer hfMm - Part 4**
> > > >
> > > > **W4. Privacy metrics: It is advisable to incorporate privacy metrics into the framework, as privacy is a primary motivation for unlearning.**
> > > >
> > > > Our original submission already included membership-inference attacks to assess privacy preservation (Section 5.4). In the revised version, we have further strengthened the evaluation by incorporating two additional privacy-oriented threat models: **unlearning inversion attacks** \[1\] and **noisy-labeler attacks** \[2\]. Together, these attacks probe different facets of residual information, ranging from prediction-level leakage (MIA) to reconstruction of hidden representations (inversion) and label-dependent vulnerability (noisy-labeler attacks). The new results and key insights are summarized below.
> > > >
> > > > * MEGU, GIF, and IDEA consistently achieve AUROC ≈ 0.50 across all attacks and datasets, demonstrating robust and stable forgetting.
> > > > * GNNDelete performs poorly on all three attacks, with AUROC often >0.80, indicating strong membership leakage and confirming that it is not safe for unlearning.
> > > > * Projector shows clear weaknesses in MIA and inversion attacks but performs reasonably under noisy-labeler attacks. This inconsistency highlights its susceptibility to geometry-based attacks, as discussed.
> > > > * GraphEraser is weak under MIA (detectable leakage), but performs well under inversion and noisy-labeler attacks. This inconsistency aligns with our earlier observation that its theoretical exactness breaks when gold is not sharded.
> > > >
> > > > ### Inversion Attack AUROC
> > > > (Closer to 0.5 is better.)
> > > > | **Method**       | **Cora**       | **Citeseer**   | **Photo**      | **Amazon-R.**  | **Roman-E.**   | **OGBN-Arxiv** |
> > > > |------------------|----------------|----------------|----------------|----------------|----------------|----------------|
> > > > | MEGU             | 0.53 ± 0.01    | 0.54 ± 0.00    | 0.53 ± 0.01    | 0.51 ± 0.00    | 0.51 ± 0.00    | 0.50 ± 0.00    |
> > > > | GIF              | 0.53 ± 0.02    | 0.54 ± 0.01    | 0.53 ± 0.01    | 0.51 ± 0.00    | 0.51 ± 0.00    | 0.50 ± 0.00    |
> > > > | IDEA             | 0.52 ± 0.01    | 0.54 ± 0.01    | 0.53 ± 0.01    | 0.51 ± 0.00    | 0.51 ± 0.00    | 0.50 ± 0.00    |
> > > > | GNNDelete        | 0.78 ± 0.05    | 0.90 ± 0.01    | 0.89 ± 0.12    | 0.95 ± 0.01    | 0.80 ± 0.01    | OOM            |
> > > > | Projector        | 0.60 ± 0.02    | 0.57 ± 0.00    | 0.53 ± 0.01    | 0.60 ± 0.01    | 0.52 ± 0.01    | 0.56 ± 0.00    |
> > > > | GraphEraser      | 0.54 ± 0.01    | 0.54 ± 0.01    | 0.53 ± 0.01    | 0.51 ± 0.00    | 0.51 ± 0.01    | 0.51 ± 0.00    |
> > > > | GUIDE            | 0.53 ± 0.00    | 0.54 ± 0.01    | 0.53 ± 0.01    | 0.51 ± 0.00    | 0.51 ± 0.00    | OOM            |
> > > >
> > > > ### Noisy-Labeler Attack AUROC
> > > > (Closer to 0.50 is better.)
> > > >
> > > > | **Method**       | **Cora**       | **Citeseer**   | **Photo**      | **Amazon-R.**  | **Roman-E.**   | **OGBN-Arxiv** |
> > > > |------------------|----------------|----------------|----------------|----------------|----------------|----------------|
> > > > | MEGU             | 0.50 ± 0.01    | 0.51 ± 0.02    | 0.50 ± 0.01    | 0.50 ± 0.00    | 0.50 ± 0.01    | 0.50 ± 0.00    |
> > > > | GIF              | 0.50 ± 0.01    | 0.51 ± 0.02    | 0.50 ± 0.01    | 0.51 ± 0.01    | 0.50 ± 0.01    | 0.50 ± 0.00    |
> > > > | IDEA             | 0.50 ± 0.01    | 0.50 ± 0.02    | 0.50 ± 0.01    | 0.50 ± 0.01    | 0.50 ± 0.01    | 0.50 ± 0.00    |
> > > > | GNNDelete        | 0.58 ± 0.03    | 0.59 ± 0.02    | 0.61 ± 0.06    | 0.81 ± 0.02    | 0.52 ± 0.02    | OOM            |
> > > > | Projector        | 0.50 ± 0.03    | 0.51 ± 0.04    | 0.50 ± 0.01    | 0.50 ± 0.00    | 0.50 ± 0.01    | 0.50 ± 0.00    |
> > > > | GraphEraser      | 0.49 ± 0.01    | 0.52 ± 0.01    | 0.50 ± 0.01    | 0.50 ± 0.00    | 0.50 ± 0.01    | 0.50 ± 0.00    |
> > > > | GUIDE            | 0.50 ± 0.01    | 0.51 ± 0.02    | 0.50 ± 0.01    | 0.51 ± 0.00    | 0.50 ± 0.01    | OOM            |
> > > >
> > > > \[1\] Jiahao Zhang, Yilong Wang, Zhiwei Zhang, Xiaorui Liu, Suhang Wang. “Unlearning Inversion Attacks for Graph Neural Networks”. WSDM 2026.
> > > >
> > > > \[2\] Zhihao Sui, Liang Hu, Jian Cao, Dora D. Liu, Usman Naseem, Zhongyuan Lai, Qi Zhang. “Recalling The Forgotten Class Memberships: Unlearned Models Can Be Noisy Labelers to Leak Privacy”. IJCAI, 2025.

---

> ### Comment · Reviewer_hfMm · 2025-11-25
>
> Thanks for the detailed rebuttal. My initial concerns have been mostly addressed, and the two additional privacy-oriented evaluations significantly enhance the benchmarking. I will therefore raise my score accordingly.

---

> > ### Author Response · Authors · 2025-11-26
> >
> > We are glad to hear that our revisions have addressed your concerns. If there are any remaining issues or additional clarifications, we can provide to further demonstrate the merits of our work, we would be very happy to discuss them.

---

### Author Response · Authors · 2025-11-23
**Summary of revisions on Benchmarking GNN Unlearning**

We thank the reviewers for their insightful and constructive feedback. We have revised the manuscript thoroughly in response to all comments. All changes in the revised version appear in **blue** text. A detailed, point-by-point response is provided in the rebuttal. Below we summarize the major improvements.

---

### 1. Expanded Experimental Coverage

- **Cognac [1] and ETR [2]** have been incorporated into the benchmark.
- We now include **two additional privacy attacks** to strengthen forgetting evaluation:
  - Inversion attacks [5]
  - Noisy-labeler attacks [6]
- The benchmark now covers multiple deletion modalities:
  - **Node unlearning** (original setting)
  - **Edge unlearning**
  - **Feature unlearning**
- We have added a **granular time breakdown**, separating preprocessing time and unlearning time. This reveals that preprocessing, which is often *not reported* in prior works, can dominate total runtime, and in many cases makes the method **slower than retraining from scratch**.
- We additionally include **certified unlearning algorithms** ScaleGun[3] and CGU[4] and discuss their scope and limitations.

### 2. Improvements in Presentation and Discussion

- We improved the exposition of forgetting quality, clarifying its importance for privacy, compliance (e.g., GDPR’s “right to erasure”), and safe deployment.
- We have added a theoretical justification for using parameter distance as a diagnostic metric. Using the Lipschitz continuity of message-passing GNNs, we now provide a lemma showing that **small deviations in parameters imply proportionally small deviations in layer activations**. This connects weight-space proximity to functional similarity in a principled manner.

---

We believe these revisions significantly enhance the clarity, completeness, and diagnostic value of the benchmark. We hope to discuss further with the reviewers in the dicussion phase and address any outstanding concerns.

---

**References:**

_\[1\] Kolipaka, Varshita, Akshit, Sinha, Debangan, Mishra, Sumit, Kumar, Arvindh, Arun, Shashwat, Goel, Ponnurangam, Kumaraguru. "A Cognac Shot To Forget Bad Memories: Corrective Unlearning for Graph Neural Networks." Proceedings of the 42nd International Conference on Machine Learning (ICML)._

*[2] Zhe-Rui Yang, Jindong Han, Chang-Dong Wang, Hao Liu. “Erase then Rectify: A Training-Free Parameter Editing Approach for Cost-Effective Graph Unlearning”. AAAI 2025.*

*[3] Lu Yi and Zhewei Wei. Scalable and certifiable graph unlearning: Overcoming the approximation error barrier, ICLR 2025*

*[4] Eli Chien, Chao Pan, and Olgica Milenkovic. Certified graph unlearning. In NeurIPS GLFrontiers Workshop, 2022.*

*[5] Jiahao Zhang, Yilong Wang, Zhiwei Zhang, Xiaorui Liu, Suhang Wang. “Unlearning Inversion Attacks for Graph Neural Networks”. WSDM 2026.*

*[6] Zhihao Sui, Liang Hu, Jian Cao, Dora D. Liu, Usman Naseem, Zhongyuan Lai, Qi Zhang. “Recalling The Forgotten Class Memberships: Unlearned Models Can Be Noisy Labelers to Leak Privacy”. IJCAI, 2025.*

---

### Author Response · Authors · 2025-12-01

Dear AC,

Since we have been informed that no further reviewer responses will be collected and that ratings have been reverted to their pre-discussion state, we would like to summarize the **verifiable trajectory** of the discussion and score revisions that occurred before the rollback.

**Scope of our work.**
We present an in-depth, diagnostic benchmarking of graph unlearning methods, evaluating whether existing approaches are ready for practical deployment.

**Initial concerns.**
Across reviews, three substantive requests emerged:
1. a more thorough evaluation of *forgetting*,
2. inclusion of additional unlearning algorithms, and
3. assessment of unlearning under *edge* and *feature* deletions in addition to node unlearning.

**Our revision.**
Our cover letter provides the summary of our rebuttal. We have also uploaded our revised manuscript fully addressing each of the major concerns above. Given the context surrounding trust issues in the reviewing process due to data leak, we respectfully encourage the AC to examine the rebuttal carefully, as it reflects a comprehensive and transparent response to all reviewer feedback.

**Reviewer reactions prior to rollback.**

- **Reviewer hfMm**
  *Initial rating: 4 → Updated rating: 6*
  (The text indicates a rating increase; the minimum possible updated rating is 6.)
  > *“Thanks for the detailed rebuttal. My initial concerns have been mostly addressed, and the two additional privacy-oriented evaluations significantly enhance the benchmarking. I will therefore raise my score accordingly.”*

- **Reviewer hgen**
  *Did not respond; rating unchanged (6).*
  We believe our rebuttal comprehensively addresses the concerns raised by this reviewer. Since the reviewer could not participate in the discussion phase due to its pre-mature closure, we kindly **request the AC to consider the substance and completeness of our rebuttal** and assess how it may have influenced the reviewer’s final rating had they had the opportunity to respond.

- **Reviewer JbxA**
  *Initial rating: 4 → Updated rating: 8*
  > *“I appreciate the very detailed rebuttal by the authors. The responses are of high quality, and cover almost all the initial weaknesses… Considering the major improvements, I am happy to bump my score to 8.”*

- **Reviewer MdZs**
  *Initial rating: 6 → Updated rating: 8*
  The reviewer interacted twice after our rebuttal: first, pointing out some technical inconsistencies in Table 1, which we corrected. Following that clarification, the rating was increased to 8.
  > *“Given the clarifications and the breadth of relevant experiments added during the rebuttal phase, I think this is a valuable contribution to the graph unlearning domain. I'm happy to increase my score to 8.”*

#### **Ratings Overview**

| Reviewer | hfMm | hgen | JbxA | MdZs |**Mean**|
|---------|------|------|------|------|-|
| **Initial** | 4 | 6 | 4 | 6 |**5.00**|
| **Updated** | 6 | 6 (no response) | 8 | 8 |**7.00**|

We thank the AC for taking time in moderating our paper submission and remain hopeful that the strengthened manuscript and the reviewers’ updated assessments will guide your final decision.

regards,

AC

---

### Meta-Review · Area_Chair_6Lzn · 2025-12-11

**Summary:**

This submission presents a systematic benchmark for GNN unlearning organized around three desiderata: efficiency (vs. retraining), utility (including alignment to a retrained “Gold” model beyond just accuracy), and forgetting (privacy-oriented tests of whether removed data no longer influences the model). The benchmark spans multiple datasets and deletion scenarios, and culminates in practical “decision-making” guidance showing that no current method satisfies all three desiderata, with different techniques trading off scalability, alignment-to-Gold, and forgetting robustness.

**Reviewer concerns before rebuttal.**
Across reviews, the main concerns were that the initial benchmark felt incomplete for a “ready for practice” claim: reviewers asked for (i) stronger and more diverse forgetting/privacy evaluations, (ii) inclusion of additional recent methods (and better coverage of the method landscape), and (iii) broader deletion modalities beyond randomized node deletions (e.g., edge/feature unlearning). Reviewers also raised issues around runtime accounting/standardization, limited certified-method comparisons, and insufficient theoretical justification for some diagnostics/metrics.


**Remaining concern(s) informing my suggested decision.**
The rebuttal and revision substantially strengthen the benchmark: they add newer attacks, expand method coverage, and broaden deletion modalities (node/edge/feature), directly addressing the biggest completeness concerns.

Overall, given the clear benchmark contribution and the strengthened experimental coverage, my suggested decision is Accept.

**Reviewer Concerns:**

I believe all the revieweer's concerns have been addressed with the comprehensive rebuttal. This is also reflected by the score raised by the reviewers.

**Reviewer Scores:**

Below is how I believe each reviewer would have adjusted their score with full discussion participation, based on their stated reactions in the thread and the post-rebuttal trajectory:

* Reviewer hfMm: 4 --> 6. The reviewer explicitly stated their initial concerns were mostly addressed and that the added privacy-oriented evaluations significantly improved the benchmark.

* Reviewer hgen: 6 --> 6 (likely unchanged).

* Reviewer JbxA: 4 --> 8. The reviewer explicitly indicated they were happy to bump to 8 after seeing the rebuttal and expanded experiments, with only a minor remaining suggestion about reorganizing appendix material into the main text.

* Reviewer MdZs: 6 --> 8. The reviewer indicated their main concern (breadth for a benchmark paper) was addressed by added experiments, and later explicitly stated they were happy to increase to 8 after clarifications.

---

### Decision · Program_Chairs · 2026-01-26

Accept (Poster)